# Solving Composable Constraints for Inverse Design Tasks

## Abstract

Inverse design tasks are an important category of problem in which we want to identify some input vector $x$ satisfying some desirable properties. In this paper we propose a mechanism for representing inequality constraints as Signed Distance Functions (SDFs). SDFs permit efficient projection of points into the solution region as well as providing a mechanism for composing constraints via boolean set operations. In this paper, we provide theoretical motivation for Signed Distance Functions (SDFs) as an implicit representation of inequality constraints. Next, we provide analysis demonstrating that SDFs can be used to efficiently project points into solution regions. Additionally, we propose two novel algorithms for computing SDFs for wide families of machine learning models. Finally, we demonstrate practical utility by performing conditional image generation using MNIST and CelebA datasets, and computational drug design using the ZINC-250K dataset. From the experimental results, we note that the composable constraints can reliably and efficiently compute solutions to complex inverse design tasks with deep learning models.

## 1 Introduction

Inverse Design and constraint satisfaction tasks are classes of problems that appear in many domains, such as computational drug design (Ingraham et al. (2023); Gómez-Bombarelli et al. (2018); Szymczak et al. (2023); Das et al. (2018); Lim et al. (2018)) and conditional image generation (Bao et al. (2017); Karras (2017); Rombach et al. (2022)). Specifically, given an objective function $f(x) : \mathbb{R}^N \to \mathbb{R}$ and constraint functions $C(x) : \mathbb{R}^N \to \{\text{False}, \text{True}\}^M$, inverse design and constrained optimization tasks correspond to finding solutions to equations 1 and 2 respectively,

$$
\begin{aligned}
& \text{find} \quad x \\
& \text{s.t.} \quad C(x) = \text{True}
\end{aligned} \tag{1}
$$

$$
\begin{aligned}
& \max_x \quad f(x) \\
& \text{s.t.} \quad C(x) = \text{True}.
\end{aligned} \tag{2}
$$

By imposing limitations on the forms that $C(x)$ and $f(x)$ can take, methods such as Linear Programming can provide strong bounds in terms of computational performance. However, the simple models allowed by such algorithms are not suitable for complex settings that require more sophisticated modelling. Conversely, machine learning models have shown success on a wide variety of complex inverse design tasks, such as drug design and conditional image generation. For example, machine learning use learned emulator models for target properties, and learned feature spaces for parameterizing designs when solving inverse design tasks. However, while the introduction of complex machine learning models provides avenues for more sophisticated modelling and design, it comes at a cost. In particular, the non-linear nature of machine learning models, coupled with high-dimensional input spaces, presents a significant challenge to efficiently solving inverse design tasks and constrained optimization problems in these settings. As a result, current methodologies suffer from a variety of limitations such as poor computational performance, limitations on the types of constraints, or requiring expensive retraining when the constraints are changed.

To address the challenges of constrained optimization and inverse design in deep learning settings, we propose a mechanism for representing and efficiently solving systems of composable constraints. In particular, by representing inequality constraints as Signed Distance Functions (SDFs), we can efficiently project points into the solution region of the inequality. Additionally, the SDF representa-

tion permits boolean operations which allow for constraints to be composed arbitrarily. For example, given a multi-output machine learning model $M(x) : \mathbb{R}^N \to \mathbb{R}^M$, the composable constraints are expressed as inequality constraints on the model outputs, as well as boolean combinations of constraints:

$$
\begin{aligned}
C(x) &= M_i(x) \geq k && \triangleright \text{ Single constraint} \\
C(x) &= (M_i(x) \geq k_i) \cap (M_j(x) \geq k_j) && \triangleright \text{ Intersection of two constraints} \\
C(x) &= (M_i(x) \geq k_i) \cup (M_j(x) \geq k_j) && \triangleright \text{ Union of two constraint} \\
C(x) &= C_1(x) \cap C_2(x) && \triangleright \text{ Composing constraints}
\end{aligned}
\tag{3}
$$

In brief, the SDF formulation provides a principled approach for defining composable constraint functions $C(x)$ (see equation 3) and efficiently solving equation 2. In the following sections, we first describe how signed distance functions (SDFs) can be used as implicit representations of inequality constraints of the form $M_i(x) \geq k$. Subsequently, we show that a chosen starting point $x_0$ can be efficiently projected into the solution region of a given constraint. Additionally, we provide two algorithms for computing the SDFs of inequality constraints for two broad families of deep learning models. Finally, we demonstrate practical utility by applying composable constraints to conditional image generation using the MNIST and CelebA datasets, as well as computational drug design using ZINC-250K.

## 2 BACKGROUND

The SDF formulation of composable constraints touches upon a variety of domains, such as constructive solid geometry, numerical optimization, and tropical geometry. The following sections aim to provide an overview of the prerequisite concepts for understanding the key theoretical aspects of composable constraints.

**Signed Distance Functions**   Signed Distance Functions (SDFs) provide a mechanism for representing volumes and solids algebraically. For an arbitrary solid, an SDF represents both membership to the volume, as well as distance from the boundary of the solid (Ricci (1973); Marschner et al. (2023)). Given a solid $\mathcal{S}$, the solid boundary $\partial S$ and a point $x \in \mathbb{R}^N$, the SDF of $S$ is defined as follows

$$
\text{SDF}_S(x) = \begin{cases} -\inf\{||x - x'||_2 \forall x' \in \partial S\} & \text{if } x \in S \\ 0 & \text{if } x \in \partial S \\ \inf\{||x - x'||_2 \forall x' \in \partial S\} & \text{if } x \notin S. \end{cases}
\tag{4}
$$

A notable property of SDFs is that volumetric operations of SDFs can be obtained via simple algebraic manipulations. In fact, several such operations are widely used in SDF-based Constructive Solid Geometry (Ricci (1973); Marschner et al. (2023)). We define unions, intersections, and negations of SDFs as:

$$
\text{SDF}_{S_1 \cup S_2}(x) \geq \min(\text{SDF}_{S_1}(x), \text{SDF}_{S_2}(x))
\tag{5}
$$

$$
\text{SDF}_{S_1 \cap S_2}(x) \geq \max(\text{SDF}_{S_1}(x), \text{SDF}_{S_2}(x))
\tag{6}
$$

$$
\text{SDF}_{\neg S_1}(x) = -\text{SDF}_{S_1}(x).
\tag{7}
$$

Unfortunately, the boolean SDF operations do not yield an exact SDF, but rather a pseudo-SDF which produces a bound for the distance, rather than an exact value (Marschner et al. (2023)). See appendix B for detailed examples of SDFs.

**Constrained Optimization**   Methods such as linear and quadratic programming impose strict restrictions on the form of $C(x)$ and $f(x)$ in order to efficiently compute solutions to equation 2. Both linear and quadratic programming restrict the constraints to linear equalities $C(X) : Ax = B$ and inequalities $C(X) : Ax \geq B$. Furthermore, linear programming requires a linear objective $f(x) : c^\top x$ (Karloff (2008)) while quadratic programming requires a quadratic objective $f(x) : \frac{1}{2}x^\top Q x + c^\top x$ (Floudas & Visweswaran (1995)). Several algorithms and corresponding computational complexity bounds exist for solving linear and quadratic programs (Vaidya (1989); Ye & Tse (1989)), allowing us to reason about performance in various settings.

Conversely, non-linear optimization makes very few assumptions on the structure of either the objective or constraint functions. While such methods are powerful and flexible, performance and convergence remains a challenge depending on the algorithm chosen (Yuan, 1991; Moritz et al., 2016). In the context of this paper, augmented Lagrangian constrained optimization algorithms are of particular interest, as they perform well in high-dimensional spaces (Nocedal & Wright, 1999). In particular, we use augmented Lagragians in order to construct a linear-time algorithm for computing SDFs in high dimensional spaces. Subsequently, we use linear and quadratic programming to construct an SDF algorithm for certain neural network architectures.

**Guided Gradient Descent**  Guided gradient descent (GGD) aims to solve inverse design tasks by explicitly inverting a predictive model. To perform GGD, we first encode the inverse design task into a differentiable loss function. A typical loss function $\mathcal{L}(x)$ might be the $L_2$ norm between the model output $M(x)$ and some desired outputs $y^*$: $\mathcal{L}(x) = ||M(x) - y^*||_2$. By performing gradient descent through the model, we optimize the objective in the input space to identify solutions to equation 1. Specifically, we apply the update rule $x_{t+1} = x_t + \nabla_{x_t} \mathcal{L}(x_t)$ until convergence.

GGD can be used for flexible post-hoc generation and can accommodate a wide variety of models and objectives. Additionally GGD has shown success in tasks such as drug design (Gómez-Bombarelli et al., 2018). However, GGD frequently produces adversarial attacks when applied naively (Goodfellow et al., 2014). In this paper, we use GGD as a baseline to which composable constraints are compared.

**Shepard interpolation Neural Networks**  Shepard Interpolation Neural Networks (SINNs) are a type of single-layer feed-forward neural network that have shown success on tasks such as image and time-series classification (Smith et al., 2018a;b; Smith & Williams, 2018; 2019; Williams, 2016). The hidden layer is parameterized by scale and bias matrices $S, B$ with output layer parameters $u$,

$$w_i = \epsilon_2 + \left( \epsilon_1 + \sum_{j=1}^{D} (S_{ij}(x_j + B_{ij}))^2 \right)^{-1}$$

$$h_i = \frac{w_i}{\sum_{i=1}^{H} w_i} \qquad y = h^\top u \tag{8}$$

where $\epsilon_1, \epsilon_2$ are small constants for numerical stability. Since the activation functions are formulated to emulate Shepard Interpolation (Williams, 2016), SINNs have several interesting geometrical properties. Notably, the output of the model passes through the set of points $(b_i, u_i)$ and as such the node weights have a concrete geometric interpretation. As a result, in section 4.1, we demonstrate how SINNs permit a linear time SDF algorithm.

## 3 COMPOSABLE CONSTRAINTS

We now can introduce the concept of composable constraints as a principled approach for solving inverse design tasks in deep-learning settings. In this section, we describe expressing and solving composable constraints assuming that a computable SDF function is available. Subsequently, in section 4, we provide two concrete algorithms for computing SDFs of wide families of models.

### 3.1 EXPRESSING CONSTRAINTS AS SIGNED DISTANCE FUNCTIONS

First, let us recall the definition of a Signed Distance Function (SDF) from equation 4. In addition to the definition, we note the intuition that an SDF is an implicit representation of a given solid volume. Given a constraint of the form $M_i(x) \geq k_i$, the region in which the constraint is satisfied is in fact a solid volume. Consequently, it is possible to represent the solution region volume implicitly using an SDF. From a more formal point of view, the level set $M_i(x) = k_i$ represents the solution region boundary and corresponds exactly to $\partial S$ from equation 4. From there, finding the closest point $x' \in \partial S, \partial S = \{x : M_i(x) = k_i\}$ yields the signed distance. The SDF representing the solution to a constraint is referred to as the constraint SDF (cSDF). See figure 1 for a visualization of the cSDF. Next, given multiple cSDFs, we can combine them algebraically using the boolean SDF CSG operations defined in equations 5, 6 and 7. Combining multiple SDFs via boolean operations will yield yet another SDF, thus allowing for arbitrary constraints to be composed.

## 3.2 Solving Composable Constraints

From the previous sections, we now have mathematical tools for expressing inequality constraints as SDFs, as well as combining them via boolean operations, allowing us to compose arbitrary constraints for inverse design tasks. However, we still need to efficiently solve the system of composed constraints. In this section, we will describe how composable constraints are solved, assuming a computable SDF is available.

First, we formulate solving the system of SDF constraints as a constrained optimization problem in equation 9. Subsequently, we make use of the properties of SDFs to efficiently find solutions to the constrained optimization problem.

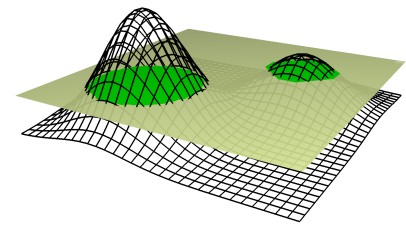

Figure 1: We define a 2D model $z = M(x, y)$, plotted using the black and white mesh. Next, the constraint $M(x, y) \geq k$ is visualized, with the constraint $k$ shown as the beige plane. The solution region $S$ is shown in green. We note the correspondence between $M(x, y) \geq k$ and a solid volume expressible by an SDF.

$$\min_{x} \quad ||x - x_0||_2$$
$$\text{s.t.} \quad C(x) = \text{True} \tag{9}$$

A useful property of SDFs is that their gradient is always a unit vector (see appendix B, eq 17). Additionally, we note that the SDF is exact in the single constraint case. Given that the value of the SDF function is the distance from the boundary, and the gradient of the SDF is the direction to the boundary, we can recover the boundary point, and thus a solution, using equation 10,

$$x = x_0 + \nabla_x \text{SDF}_s(x_0) \cdot \text{SDF}_s(x_0). \tag{10}$$

However, equation 10 is only valid if the SDF is exact. As described in section 2, the boolean operations used to combine the cSDFs result in a pseudo SDF and as such, the projection rule in equation 10 is not guaranteed to produce a point on the solution volume boundary. Instead, we can take multiple steps until a solution is found. Given a step size $\alpha$, we incrementally take steps in the direction of the SDF using the update rule

$$x_{t+1} = x_t + \alpha \nabla_x \text{SDF}_s(x_t) \cdot \text{SDF}_s(x_t). \tag{11}$$

Another consideration when solving the constraint queries is that the boolean SDF operations defined in equations 5, 6, and 7 consist of discontinuous min and max operations. The discontinuous nature of the min and max functions leads to inefficient updates steps, since the gradient will only account for a single constraint at a time. Consequently, we replace the hard min and max operations with the Log-Exp-Sum smooth approximation. In the case of two SDFs with similar magnitude, the Log-Exp-Sum function defined in equation 12 allows for the gradient to smoothly interpolate between the two SDFs, allowing for faster and more robust convergence.

$$\nu = \frac{1}{\beta} \log \left( e^{\beta a} + e^{\beta b} \right). \tag{12}$$

In conclusion, we now have a method for solving composable constraints assuming that an SDF algorithm is given. In section 4, we describe novels algorithms for computing SDFs in a wide range of settings.

## 4 Algorithms for Computing Signed Distance Functions

As described in section 3, we can express and solve complex systems of constraints using SDFs as an implicit representation of the solution region. However, this approach requires that the SDF can be computed for non-linear models in high-dimensional spaces. Unfortunately, current methods for computing SDFs are either overly restrictive in the types of models which can be used, or are intractable in high-dimensional spaces (Molchanov et al., 2010; Lu et al., 2018; Fuhrmann et al., 2015; Wu & Kobbelt, 2003; Huang & Wang, 2010; Ottaviani & Sodomaco, 2020). Consequently, novel SDF algorithms are required. In this section, we show that by imposing mild requirements on

the model form, that it is possible to derive efficient algorithms for computing the cSDFs. Notably, we present a linear-time algorithm for computing cSDFs for a wide class of smooth functions, as well as a heuristic-search based algorithm for computing cSDFs of neural networks with arbitrary continuous piece-wise linear activation functions.

### 4.1 Signed Distance Functions Smooth Asymptotic functions

To compute the SDF, we need to solve equation 9. To this end, if we can efficiently enumerate all potential solutions regions, then we can then identify the nearest point on the solution boundary, thus computing the SDF. Unfortunately, enumerating the solution regions remains a difficult problem. However, if a function is continuously differentiable and asymptotic such that $\lim_{||x||_2 \to \infty} M(x) = c, |c| < \infty$, then each solution region to the constraint $M_i(x) \geq k, k > c$ or $M_i(x) \leq k, k < c$ must necessarily contain at least one extrema. Consequently, by enumerating the extrema of such a function, we can enumerate the single-constraint solution regions and thus compute the SDF. We formalize this intuition in Theorem 1 and appendix C.

**Theorem 1** Let $M : \mathbb{R}^n \to \mathbb{R}$, such that the image of $M$ is continuous, continuously differentiable and asymptotic such that $\lim_{||x||_2 \to \infty} M(x) = c$ and $a \leq M(x) \leq b, a, b \in \mathbb{R}, \forall x \in \mathbb{R}^N$. Let $k \in \mathbb{R}$ where $c \neq k$ and $a \leq k \leq b$. Then under the constraint $M(x) \leq k$ or $M(x) \geq k$, a search algorithm need only search among the critical points and local extrema of $M$ to compute the Signed Distance Function.

---

**Algorithm 1** Algorithm 1 : Linear-Time SINN cSDF Algorithm

---

**Require:** $M_i : \mathbb{R}^n \to \mathbb{R}^n$, $x_0 \in \mathbb{R}^n$ and $c \in \mathbb{R}$
    distances $\leftarrow []$
    region $\leftarrow \text{sign}(M_i(x_0) - c)$          ▷ Keep track if $x$ satisfies constraint
    $E \leftarrow \{-b_i | \forall b_i \in b\}$          ▷ Extract extrema from SINN params
    **for** $e$ in $E$ **do**
        **if** $\text{sign}(M_i(e) - c) \neq$ region
            $p_1 \leftarrow \text{AugmentedLagrangian}(e, c, M_i)$     ▷ Compute nearest point on boundary
            $d \leftarrow ||x_0 - p_1||_2$
            distances.append($d, x_0 - p_1$)
    **end for**
    **return** region $\cdot$ min(distances)          ▷ Returns tuple of distance and gradient

---

Theorem 1 provides an avenue for implementing an efficient cSDF algorithm for a wide family of models. In fact, the requirement that the function is continuously differentiable and asymptotic is quite permissive and can be easily satisfied by a variety of machine learning models. However, the theorem also requires that the extrema of the function can be efficiently enumerated. For most models, enumerating the extrema poses a difficult challenge. To address this challenge, we make use of Shepard Interpolation Neural Networks (SINNs) as defined in section 2. First, we show that SINNs are continuously differentiable and asymptotic (see appendix C). Next, we note that SINNs permit a convenient mechanism for enumerating the extrema. In fact, the entries of the learnable parameter matrix $B$ represent the coordinates of the extrema of the output of the corresponding SINN model (see appendix C).

Consequently, SINNs satisfy the requirements of theorem 1, as well as providing an efficient mechanism for enumerating the extrema. Thus, we can apply algorithm 1 to SINNs in order to efficiently compute the cSDF. The full details are given in algorithm 1 and appendix C, with computational complexity analysis in M.

### 4.2 Signed Distance Functions of Piece-wise Linear Neural Networks

Another model class of interest are ReLU networks. Unfortunately, ReLU networks do not satisfy the prerequisites of theorem 1. Consequently, a different approach must be taken to compute cSDFs of ReLU networks. In this section, we will describe an algorithm for computing cSDFs of neural networks with continuous piece-wise linear activation functions.


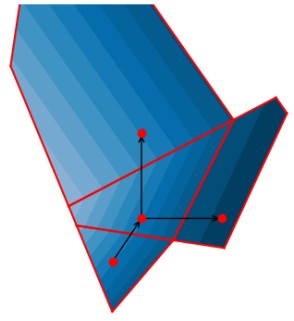

(a) Visualization of the piece-wise linear domains of a 2D ReLU network $z = \text{ReLU}(x, y)$. The boundaries of the linear domains are highlighted in red.

(b) Example of a linear domain-wise local search for computing the SDF of a ReLU network

Figure 2: Visualization of ReLU SDF algorithm.

Given a neural network with a continuous piece-wise linear activation function $f(x)$, the output of the neural network is itself piece-wise linear and continuous. Thus, the ReLU network is separable into discrete linear domains (Zhang et al. (2018)). Subsequently, given a linear domain, it is possible to solve a quadratic programming sub-problem to find the closest solution within the linear domain. By enumerating all domains, we could then compute the cSDF of the neural network. However, recent results from tropical geometry indicate that the number of linear domains grows combinatorially with the network depth, and as such it is intractable to enumerate all the domains (Zhang et al. (2018)). However, in order to compute the SDF we need only find the nearest solution, rather than enumerating all possible solutions. Consequently, performing a local search would allow the search space to be greatly reduced and thus improve the performance of a cSDF algorithm. Thus, if we can enumerate neighbouring linear domains of a given starting point, we could attempt to perform a local search such as Breadth-First Search (BFS).

In particular, we need two operations for a piece-wise linear SDF algorithm: 1) given a starting point $x_0$, efficiently identify the linear domain to which it belongs and 2) given a linear domain, find adjacent linear domains. Using these two operations, we can discretize the piece-wise linear neural network into a graph of connected linear domains, which we then traverse to find the closest solution satisfying the given constraint. For example, 2 demonstrates a 2D example of a linear-domain based local search. Furthermore, see appendix D for the derivation of both operations as well as the full ReLU SDF algorithm, and computational complexity analysis in M.

## 5 EXPERIMENTAL RESULTS

With the theoretical foundations established, we can attempt to apply composable constraints to inverse design tasks on real datasets. Notably, we applied composable constraints to image generation using MNIST (Deng, 2012) and CelebA (Karras, 2017; Liu et al., 2015), and small-molecule design using ZINC (Gómez-Bombarelli et al., 2018; Irwin & Shoichet, 2005). For the various datasets, we first train predictive models. Next, we select random training points as the initialization $x_0$, then solve the constraints using equations 10 and 11. The solution is then validated by an oracle model to compute the agreement rate.

Additionally, when dealing with regression tasks (such as ZINC), we implement a confidence-based constraint threshold adjustment. In particular, given the residuals of the predictive model on the training set, we can compute the standard deviation of the residuals as $\sigma$. From there, given a constraint threshold $k$, we choose a confidence parameter $\alpha$ to adjust how much mass of predictive distribution $\mathcal{N}(M(x), \sigma)$ is expected to satisfy the constraint inequality, following equation 13,

$$C(x) = M(x) \geq k + \alpha\sigma. \tag{13}$$

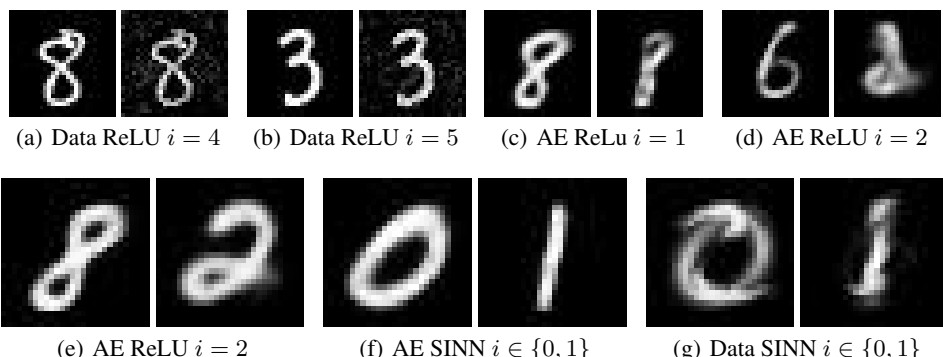

(a) Data ReLU $i = 4$   (b) Data ReLU $i = 5$   (c) AE ReLu $i = 1$   (d) AE ReLU $i = 2$

(e) AE ReLU $i = 2$   (f) AE SINN $i \in \{0, 1\}$   (g) Data SINN $i \in \{0, 1\}$

Figure 3: Generated MNIST Digits using data space and auto-encoder (AE) based ReLU and SINN classifiers. In figures a-e, we present the initialization point as well as the generated sample. In figures f and g, we present 4 generated samples using data and AE SINN models without showing the initialization points.

**Guided Gradient Descent Baseline:**   As a baseline, we ran guided gradient descent with an $L_2$ norm objective, using the same models and thresholds as in the composable constraints settings.

**Variational Auto-Encoder:**   In certain experiments, we introduce a Variational Auto-Encoder (VAE) to perform dimensionality reduction. We observe that this improves computational performance, as well as generative quality.

**MNIST:**   In total, one auto-encoder, four classifiers and an oracle were trained on the MNIST dataset. We trained SINN and ReLU classifiers on both the embeddings generated by the auto-encoder, as well as on the raw pixel features. Finally, we fine-tuned a pre-trained Resnet18 (He et al., 2016) classifier for use as an oracle to validate the results generated by the composable constraint. See appendix F for the full model architectures and training details. The MNIST constraints is used to assign high probability to a target class $i$ and is given by:

$$C(x) = M_i(x) \geq 0.9. \tag{14}$$

Table 1: Agreement Rates by task (raw input or auto-encoded (AE)), model and methodology for MNIST dataset

| Method | SINN | | ReLU | |
|---|---|---|---|---|
| | Raw | AE | Raw | AE |
| Proposed | $98.7 \pm 0.3$ | $100.0 \pm 0.0$ | $10.5 \pm 1.0$ | $43.9 \pm 1.5$ |
| GGD | $97.4 \pm 0.5$ | $97.5 \pm 0.5$ | $10.0 \pm 0.9$ | $21.7 \pm 1.3$ |

**CelebA:**   In total, one oracle model, one auto-encoder and two classifiers were trained. First, we train an auto-encoder on the latent representation produced by TinyVAE (Bohan, 2024). Stacking two auto-encoders allows for a much smaller latent space, making the SDF algorithms more computationally efficient. Next, we train SINN and ReLU classifier on the latent space produced by the second auto-encoder. Finally, we fine-tune a pre-trained ResNet18-based classifier on the original CelebA images for use as an oracle model. See appendix C for full model architecture and training details. The constraints for the single and multi-constraint cases are given in equation 15:

$$
\begin{aligned}
C(x) &= M_{\text{label}_i}(x) \geq 0.9 &&\triangleright \text{ single constraint} \\
C(x) &= M_{\text{Black Hair}} \geq 0.9 \cap M_{\text{Male}}(x) \geq 0.9 &&\triangleright \text{ multi constraint}
\end{aligned}
\tag{15}
$$

Table 2: Agreement Rates by task (single or multi-constraint), model and methodology for CelebA dataset

| Method | SINN | | ReLU | |
|---|---|---|---|---|
| | Single | Multi | Single | Multi |
| Proposed | $52.0 \pm 2.7$ | $60.9 \pm 10.1$ | $54.5 \pm 2.1$ | $46.2 \pm 6.9$ |
| GGD | $5.6 \pm 0.7$ | $0.0 \pm 0.0$ | $1.6 \pm 0.4$ | $0.0 \pm 0.0$ |

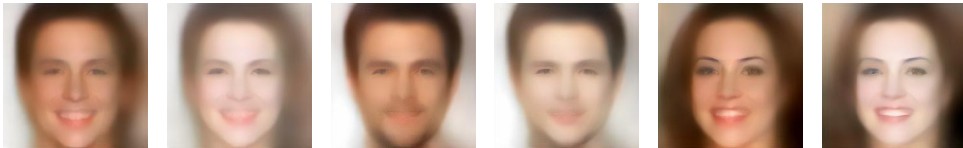

Figure 4: Visualization of initialization and generated samples using a SINN classifier and a constraint $C(x) = M_{\text{Pale Skin}}(x) \geq 0.9$

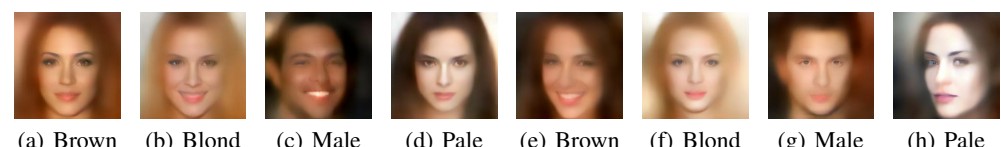

(a) Brown (b) Blond (c) Male (d) Pale (e) Brown (f) Blond (g) Male (h) Pale

Figure 5: Sample generated CelebA samples using SINN and ReLU models for Brown Hair (Brown), Blonde Hair (Blond), Male and Pale Skin (Pale). Figures a-d generated using a ReLU classifier, Figures e-h generated using an SINN classifier.

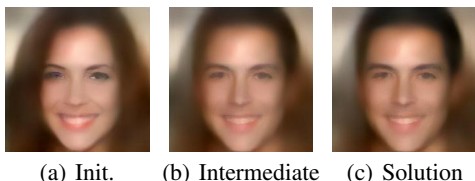

(a) Init. (b) Intermediate (c) Solution

Figure 6: Visualization of generated samples for the two class setting of "Black Hair" and "Male" using equation 8. Starting point (A), an intermediate iteration step (B) and the final generated sample (C) are show.

**ZINC-250k:** ReLU and SINN regressors were trained to predict the QED, logP and SAS given embeddings produced by ChemVAE (Gómez-Bombarelli et al., 2018). The Gaussian Process (GP) oracle provided by ChemVAE was used to validate the properties given a latent vector. Additionally, the latent vectors are decoded, with the QED and SAS computed analytically from the decoded chemical structure using RDKit (Landrum (2022); Ertl & Schuffenhauer (2009)). See appendix C for the full architecture and training details. The ZINC constraints are given in equation 16,

$$
\begin{aligned}
C_{\text{QED}}(x) &= M_{\text{QED}}(x) \geq 0.7 + 1.5\sigma_{\text{QED}} &&\triangleright \text{ QED constraint} \\
C_{\text{SAS}}(x) &= M_{\text{SAS}}(x) \geq 3.35 + 1.5\sigma_{\text{SAS}} &&\triangleright \text{ SAS constraint} \\
C_{\text{multi}}(x) &= C_{\text{QED}}(x) \cap C_{\text{SAS}}(x) &&\triangleright \text{ Multi constraint.}
\end{aligned}
\tag{16}
$$

**Result Summary:** First, in order to validate the methodology, we applied composable constraints to conditional image generation using the MNIST dataset. We report the agreement rates in table 1, accuracy and runtime metrics in appendix H, and sample generations in figure 3. We note that we can successfully generate images with a high agreement rate in both latent space and data space when using the SINN model. However, the ReLU model is liable to generate adversarial samples in

Table 3: Agreement Rates by task (single or multi-constraint), model and methodology for ZINC dataset when using Latent Oracle

| Method | SINN | | ReLU | |
| --- | --- | --- | --- | --- |
| | Single | Multi | Single | Multi |
| Proposed | $90.0 \pm 9.0$ | $93.0 \pm 14.2$ | $59.0 \pm 6.8$ | $20.0 \pm 3.0$ |
| GGD | $85.5 \pm 6.1$ | $73.0 \pm 7.3$ | $64.5 \pm 5.1$ | $32.0 \pm 3.2$ |

Table 4: Agreement Rates by task (single or multi-constraint), model and methodology for ZINC dataset when using Analytical Oracle

| Method | SINN | | ReLU | |
| --- | --- | --- | --- | --- |
| | Single | Multi | Single | Multi |
| Proposed | $43.7 \pm 1.4$ | $15.2 \pm 0.8$ | $39.6 \pm 1.3$ | $12.1 \pm 0.6$ |
| GGD | $44.8 \pm 1.1$ | $15.5 \pm 0.6$ | $43.4 \pm 1.1$ | $14.3 \pm 0.5$ |

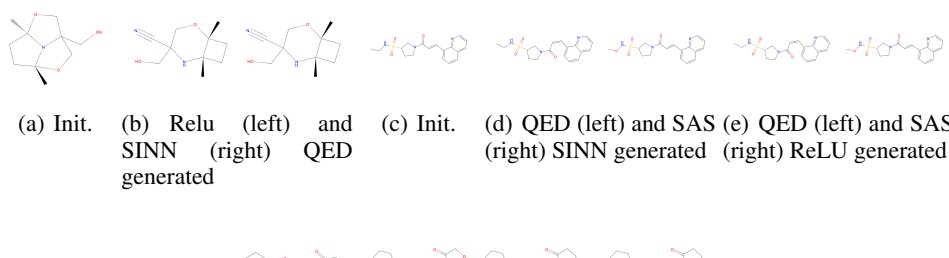

(a) Init.    (b) Relu (left) and SINN (right) QED generated    (c) Init.    (d) QED (left) and SAS (right) SINN generated    (e) QED (left) and SAS (right) ReLU generated

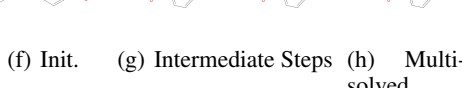

(f) Init.    (g) Intermediate Steps    (h) Multi-solved

Figure 7: Sample generated chemical structures for the ZINC dataset when performing single and multi-constraint generation. Note that the chemical properties are optimized while introducing minimal changes to the chemical structure.

both data space and latent space, despite having a much higher classification accuracy than the SINN model (see appendix H for classification accuracy). Furthermore, we note that the introduction of a VAE improves the computational performance of both SDF algorithms by a significant amount, in addition to improving the quality of the ReLU generated samples. Additionally, composable constraints outperform GGD across all MNIST sub-experiments. From this, we can conclude that composable constraints function as intended in both data and latent space, outperforms standard GGD in simple settings, and provides flexible post-hoc conditional generation while making few assumptions on the structure of the models or input space.

Next, we apply composable constraint to a more complex conditional image generation task using CelebA. Notably, CelebA provides much richer data than MNIST, in addition to supporting multi-class conditional generation. We report the agreement rates in table 2, the accuracy and runtime metrics in appendix G and sample generations in figures 4, 5 and 6. From the agreement rates in table 2, we note that composable constraints can perform conditional generation with high agreement rates in both single and multi-objective generation. Additionally, we note that composable constraints outperform GGD significantly, due to the later generating adversarial samples. Furthermore, from the example generated samples, we can see that composable constraint can perform sophisticated semantic operations despite very little assumptions made by the framework. In particular, figure 4 demonstrate that composable constraints can modify the content of the image via semantic operations that produce the desired label, while leaving other distinguishing features intact. Fur-

thermore, figure 6 demonstrates modifications to both the hair color and gender labels of the image, while maintaining facial structure and expression intact. This demonstrates the power and flexibility of composable constraints, as we can perform sophisticated conditional generation entirely post-hoc and with very little assumptions on the structure of the input or latent space, allowing for widespread application.

Finally, we apply composable constraints to computational drug design. This task was chosen to demonstrate that the proposed methodology is not only applicable to image generation tasks, but rather is a principled framework that can be applied to arbitrary settings so long as a suitable predictive model is available. In the case of the ZINC experiments, we report the agreement rate before decoding using a latent-space oracle, as well as after decoding using an analytical oracle. We report the agreement rates with the latent and analytical oracles in tables 3 and 4, as well as runtime and accuracy metrics in appendix I, and sample generations in figure 7. From the agreement rates, we see that composable constraints achieve extremely high agreement rates with the latent oracle in both single and multi-constraint generation. However, the agreement rate is much lower when computed with the analytical oracle. Furthermore, composable constraints perform on-par with GGD in this setting. This is an expected result, as ChemVAE (Gómez-Bombarelli et al. (2018)) is jointly trained with a predictive model, and thus the latent space will have a smooth structure that is amenable to GGD. However, this further validates composable constraints as an approach, as it performs on-par with GGD in a conventional setting for GGD, while providing composability at no extra modelling cost for either the generative model or the predictive model. In fact, in order to optimize a new molecular property not available in the original training set, GGD would require jointly retraining the VAE and regressor on the new property, while composable constraints would allow us to simply add a new predictive model for the property of interest, and use the current models as-is.

## 6 RELATED WORKS

In recent works, a variety of generative models, such as Variational Auto-encoders (VAEs), Generative Adversarial Networks (GANs) and diffusion models have achieve impressive results in a variety of inverse design tasks such as conditional image generation Karras (2017); Rombach et al. (2022) and drug discovery Gómez-Bombarelli et al. (2018); Szymczak et al. (2023). Notably, VAEs can perform unconditional (Blei et al. (2017)) or conditional generation (Lim et al. (2018)). While not fundamental to composable constraints, we make use of VAEs for dimensionality reduction to improve computational performance and sample quality. Similarly, GANs are typically used for conditional (Mirza & Osindero (2014)) or unconditional (Karras (2017)) image generation. However, unlike VAEs, GANs do not produce a semantic latent space and thus are not applicable to composable constraints. Alternatively, diffusion models are a powerful class of models used in image generation and chemical design (Rombach et al. (2022); Bohan (2024); Ho et al. (2020); Ingraham et al. (2023)). Unfortunately, diffusion models do not produce semantic latent spaces, and thus are not easily combined with composable constraints.

## 7 CONCLUSION

In this paper, we propose a novel method for constructing and solving composable constraints on deep learning models using Signed Distance Functions (SDFs). We provide theoretical motivation for the approach, two novel algorithms for computing SDFs in high-dimensional spaces, and empirically validate the proposed methodology on conditional image generation and computational drug design tasks. Overall, composable constraints provide a principled framework for flexible, post-hoc conditional generation.

**Limitations** First, we note that composable constraints require a computable SDF function, limiting the applicability of this method to model families with known SDF functions. Additionally, the algorithms for computing the SDF are more complex and computationally intensive than GGD, particularily for the ReLU SDF algorithm, and thus may not be suitable for very large models. See appendices G, H and I for algorithm runtime results. However, solutions are typically computed in a few seconds, and thus are fast enough to be useful in practice. Finally, composable constraints are novel, and thus are poorly characterized. In particular, the impact of the predictive model form on the agreement rate is not well understood and would require further work to fully characterize.

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

# A APPENDIX

Table 5: Important Variables and relevant comments.

| Variable Notation | Description |
|---|---|
| $S$ | A region with a boundary and interior volume |
| $\partial S$ | The boundary of a solid |
| $k$ | A scalar value typically used for a constraint threshold |
| $M(X)$ | An arbitrary machine learning model |
| $M_i(X)$ | i$^{\text{th}}$ output of a machine learning model |
| $\text{SINN}(x)$ | A Shepard Interpolation Neural Network |
| $\text{ReLU}(x)$ | A fully-connected ReLU network |
| $\text{SDF}_s(x)$ | Signed Distance Function of the solid $S$ |
| $\|\cdot\|_2$ | The $L_2$-norm of a vector |
| $\cap$ | Boolean intersection |
| $\cup$ | Boolean union |
| $\neg$ | Boolean negation |
| $\nabla_x$ | The Jacobian with respect to $x$ |
| $\mathcal{N}(\mu,\sigma)$ | Normal distribution of mean $\mu$ and variance $\sigma^2$ |
| $k^{\pm 1}$ | Multiplication or division by $k$ |
| $\sigma$ | A standard deviation |
| $\tilde{v}$ | Transformed value of a variable $v$ |
| $\nu$ | Approximate min/max value |
| $\alpha$ | Numerical hyper-parameter |
| $\beta$ | Numerical hyper-parameter |
| $x$ | Input vector |
| $x_0$ | Starting point |
| $y$ | Output vector |
| $y^*$ | Target vector |
| $z$ | Latent vector |

# B    EXAMPLES OF SIGNED DISTANCE FUNCTIONS

## B.1    GRADIENT OF SIGNED DISTANCE FUNCTION

Given a starting point $x$, and the nearest boundary point $x'$, we show that the gradient of the SDF is a unit vector.

$$
\begin{aligned}
\text{SDF}(x) &= ||x - x'||_2 \\
&= \sqrt{(x - x')^\top (x - x')} \\
\nabla_x \text{SDF}(x) &= \nabla_x \sqrt{(x - x')^\top (x - x')} \\
&= \frac{x - x'}{||x - x'||_2} \\
\therefore ||\nabla_x \text{SDF}(x)||_2 &= 1
\end{aligned}
\tag{17}
$$

## B.2    BOOLEAN SIGNED DISTANCE FUNCTION OPERATIONS

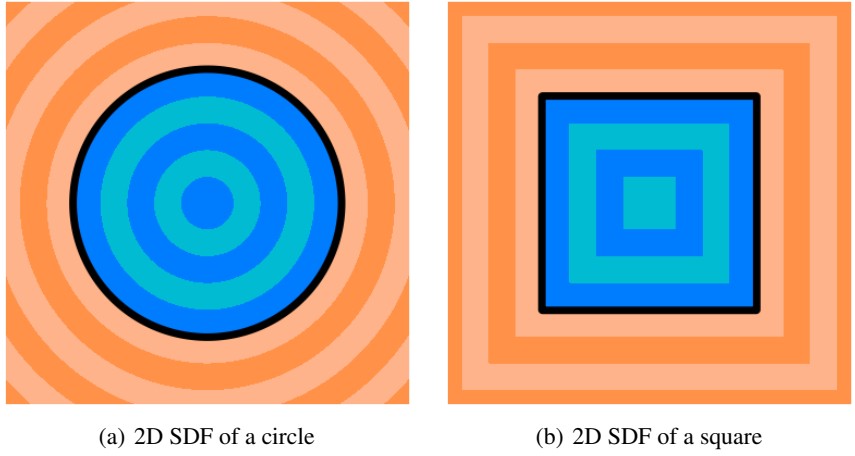

(a) 2D SDF of a circle          (b) 2D SDF of a square

Figure 8: Example Signed Distance Functions (SDF) of a square and a circle. The solid boundary $\partial S$ is indicated by a thick black line, the interior of the solid $S$ is indicated in blue, while the exterior is indicated in orange. Bands of a single colour indicate the iso-level of the SDF distance.

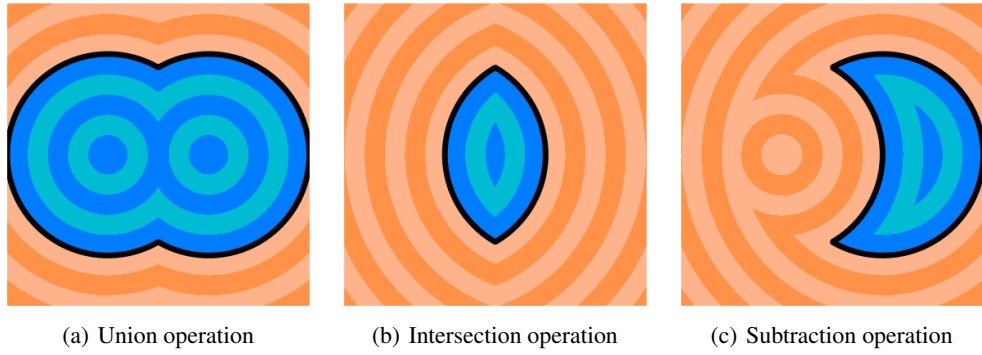

(a) Union operation          (b) Intersection operation          (c) Subtraction operation

Figure 9: Example of applying equations 5, 6 and 7 to two circle SDFs .

### B.3 EXAMPLES OF SIGNED DISTANCE FUNCTIONS IN 2 DIMENSIONS

As an example, we can examine an $N$-sphere of radius $r$ centered on the origin. Both the $N$-sphere boundary and the SDF of the $N$-sphere have known closed form equations:

$N$-**sphere boundary:**

$$r^N = \sum_{i=1}^{N} x_{\text{i}}^2 \tag{18}$$

$N$-**sphere SDF:**

$$SDF_{\text{Sphere}}(x) = \sqrt{\sum_{i=1}^{N} x_{\text{i}}^2} - r \tag{19}$$

In 2 dimensions, we can visualize the 2-sphere alongside it's equivalent SDF:

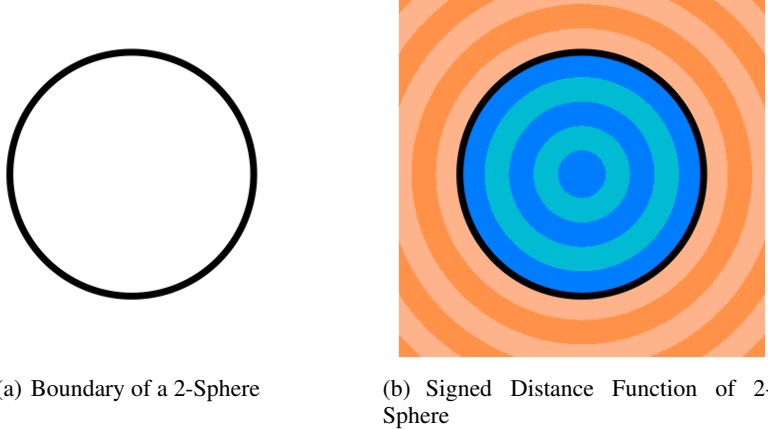

(a) Boundary of a 2-Sphere    (b) Signed Distance Function of 2-Sphere

Figure 10: 2-Sphere and corresponding Signed Distance Function

### B.4 EXAMPLE FUNCTION WITH CONSTRAINT BOUNDARY IN 2 DIMENSIONS

A simple 2D example is given to illustrate the interaction between a given function $f(x)$, the constraint boundary $f(x) = k$, the constraint solution region $S$ and the SDF of the solution region $SDF_S(x)$. More specifically, Mishra's bird function was chosen from a list of sample optimization functions Mishra (2006). For the example, a constraint of $f(x) \geq 25$ was chosen to provide a visually interesting constraint boundary.

$$f(x) = sin(x_2)e^{(1-cos(x_1))^2} + cos(x_1)e^{(1-sin(x_2))^2} + (x_1 - x_2)^2 \tag{20}$$

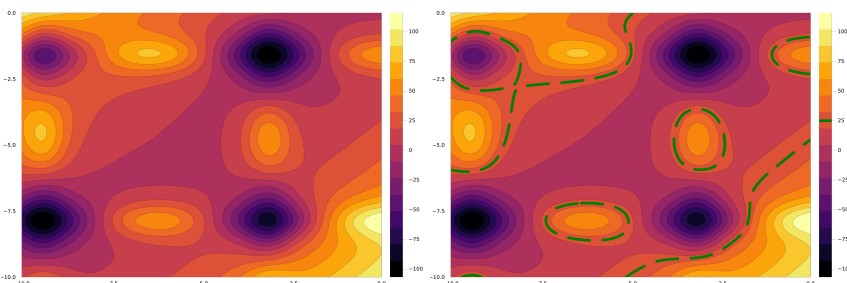

(a) Mishra's bird function Mishra (2006).   (b) Constraint boundary $f(x) = 25$ in green.

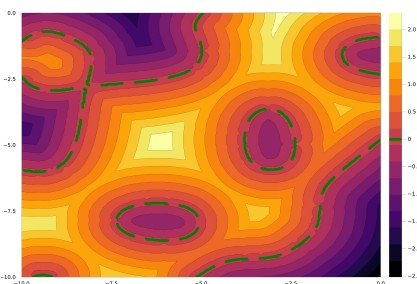

(c) Signed Distance Function of constraint set.

Figure 11: Visualization of Mishra's bird function with a constraint of $f(x) \geq 25$

## C  SINN SDF Algorithm Proof

The proof in this appendix aims to support Theorem 1 as well as provide confidence in the theoretical correctness of the SDF algorithm. This proof consists of a general proof for computing SDFs of functions meeting the prerequisite conditions, as well as a proof that SINN satisfy the prerequisite conditions.

**Proof of Theorem 1**  Let $M : \mathbb{R}^n \to \mathbb{R}$, such that the image of $M$ is continuous, continuously differentiable and asymptotic such that $\lim\limits_{||x||_2 \to \infty} M(x) = c$ and $a \leq M(x) \leq b, a, b \in \mathbb{R}, \forall x \in \mathbb{R}^N$. Let $k \in \mathbb{R}$ where $c \neq k$ and $a \leq k \leq b$. Then under the constraint $M(x) \leq k$ or $M(x) \geq k$, a search algorithm need only search among the critical points and local extrema of $M$ to compute the Signed Distance Function. See appendix F for proof.

For some constraint threshold $k$ such that $a \leq k \leq b \in \mathbb{R}$ where $M(x) \leq k$ s.t. $c > k$ or $M(x) \geq k$ s.t. $c < k$, then the boundary $M(x) = k$ is bounded by $k - \epsilon \leq M(x) \leq k + \epsilon$ for $0 < \epsilon$. Consequently, since $M(x)$ is bounded by two planes $k - \epsilon, k + \epsilon$ and $M(x)$ is asymptotic, then the boundary $M(x) = k$ must enclose a finite region. Given that the solution region of the constraint is thus bounded and finite, we can conclude by Rolle's theorem that each finite and bounded region must contain at least one critical point or local extrema . Thus, all solution regions of interest can be found by searching among the critical points of $M(x)$.

**Proof that SINNs are bounded**  In order to use SINNs for our efficient SDF algorithm, we must show that SINNs meet the conditions laid out in theorem 1. First, given the definition of the SINN in equation 8, we can analyze the inverse distance weight:

$$w_i(x) = \epsilon_2 + \frac{1}{\epsilon_1 + \sum\limits_{j=1}^{d} (s_{ij} \cdot (x_j + b_{ij}))^2} \tag{21}$$

Given equation 21, we can see that the weight tends asymptotically to $\epsilon_2$:

$$\lim_{||x||_2 \to \infty} w_i(x) = \epsilon_2 + \frac{1}{\epsilon + \infty} = \epsilon_2 \tag{22}$$

Consequently, the SINN model $M$ will tend asymptotically towards an average value $\bar{u}$:

$$\lim_{||x||_2 \to \infty} M(x) = \frac{\sum\limits_{i=1}^{m} u_i \cdot \epsilon_2}{m \cdot \epsilon_2} = \bar{u} \tag{23}$$

Next, we show that the learnable parameters $B$ correspond to the extrema of the SINN outputs:

$$
\begin{aligned}
y(x) &= \frac{\sum_i u_i w_i(x)}{\sum_i w_i(x)} \\
\nabla_x y(x) &= \frac{\nabla_x \left( \sum_i u_i w_i(x) \right) \cdot \sum_i w_i(x) - \sum_i u_i w_i(x) \cdot \nabla_x \left( \sum_i w_i(x) \right)}{\left( \sum_i w_i(x) \right)^2} \\
\lim_{x \to B_k} \nabla_x y(x) &= \frac{u_k \cdot \nabla_x w_k(x) \cdot w_k(x) - u_k \cdot w_k(x) \cdot \nabla_x w_k(x)}{\left( \sum_{i=1} w_i \right)^2} \\
\lim_{x \to B_k} \nabla_x y(x) &= 0 \\
\therefore \nabla_x y(x) &= 0 \text{ for } x = B_k
\end{aligned}
\tag{24}
$$

From the previous statements and the definition of the SINN, we know that the SINN is bounded by $k_1, k_2 \in \mathbb{R}$ such that $k_1 \leq M(X) \leq k_2$ and that $\lim\limits_{||x||_2 \to \infty} M(x) = \bar{u}$. Additionally, from equation 24, we know that the extrema of the SINN can be efficiently retrieved given the model weights. Consequently, SINNs meet all of the requirements for Theorem 1 to apply.

**Proof of Algorithm 1** The intuition for algorithm 1 is that we enumerate the extrema of the model which satisfy the given the constraint $M(x) \geq k$ or $M(x) \leq k$, and then perform constrained optimization to find the closest point on the boundary. Then, given the minima on each boundary, we can select the global minima and thus compute the cSDF.

In algorithm 1, augmented Lagrangians are used to perform the constrained optimization to compute the SDF. In fact, augmented Lagrangians can be used to solve constrained optimization under equality constraints of the form $M(x) = k$ or inequality constraints of the form $M(x) \geq k$, using only the Jacobian, allowing for linear scaling.

**Assumption** It is not known in general if the global minima of the solutions to equation 9 is always found among the local minima $p_1$ of solutions generated by constrained optimization among the extrema $e$. However, a SINN of a single node yields a flat surface, and any additional local curvature of the hyper surface is the result of the additional nodes. As such, it may be possible to construct a proof by induction that all local curvature is due to a combination of nodes, and thus any local minima would occur in the neighbourhood of a $p_1$. On the other hand, in the case that the statement does not hold true, the SDF algorithm will produce an upper bound pseudo-SDF. Computing an upper bound is still acceptable in practice, since a solution to the qSDF can still be found via gradient descent as evidenced by the results of the case studies.

**Corollary** For the case that $M(x) \geq k$ $s.t.$ $c \geq k$ or $M(x) \leq k$ $s.t.$ $c \leq k$, the constraint crosses the asymptote of $M(x)$ and thus the solution region will be infinite. However, given the definition of the SDFs, the constraints $M(x) \leq k$ and $M(x) \geq k$ share the same boundary $\partial S$. Thus, if the constraint crosses the asymptote, we can substitute with $M(x) \leq k = \neg(M(x) \geq k)$. The substitution produces the same SDF, while avoiding issues with infinite solution regions, allowing theorem 1 to still apply.

For any point $p$ that does not satisfy the constraint, all solution regions can be found by searching among the critical points of the SINN. However, for initial points $p$ that are within the solution region, we must search points outside of the solution region to identify the boundary. In order to identify the boundary from within a region, we can construct an additional constraint to identify a complement set of solutions from the original constraint, and prove that there must exist critical points outside of the solution region and would thus be able to identify the solution region boundary from within via the same algorithm. For a constraint $M(x) \leq k$ $s.t.$ $c < k$, we can construct an alternative constraint $M(x) \geq k + \epsilon$ $s.t.$ $0 < \epsilon$. Similarly, for the case $M(x) \geq k$ $s.t.$ $k \leq c$ we can construct the alternative constraint $M(x) \leq k + \epsilon$ $s.t.$ $0 < \epsilon$. From these alternative constraints, we can demonstrate that for any constraint, that there exists a complement solution region, such that at least one critical point will exist outside of the initial solution region. Thus the algorithm must only search among the critical points of the function to identify the boundary and thus the cSDF, even if the starting point is initially within the solution region.

In conclusion, all the solution regions are guaranteed to contain at least one critical point and thus all solution regions will be found when searching among the critical points of a model meeting the requisite properties. Next, the local solutions $p_1$ are computed by augmented Lagrangian for each extrema $e_i$ in order to produce the set of all of the solutions to equation 9. Thus, all solutions will be found and algorithm 1 will correctly compute the cSDF.

# D ReLU SDF Algorithm Proof

In order to identify the linear domain to which a point $x_0$ belongs, we first enumerate and number the piece-wise linear domains of the activation function, as shown in figure 12. Subsequently, we note that for some given input point $x_0$, that each neuron in the neural network will activate within one of the linear domains of the activation function. Thus, we can take note of the domain in which each neuron activation occurs, thus labelling neuron. The collection of activation domain labels for the entire network is called the activation configuration. For a given activation configuration, the model is linear, and thus there is an equivalent linear model applicable strictly within the domain of the given activation configuration. Additionally, the network itself will behave non-linearly if and only if the activation configuration changes. Consequently, we can find systems of linear inequalities which define the domain of the given activation configuration.

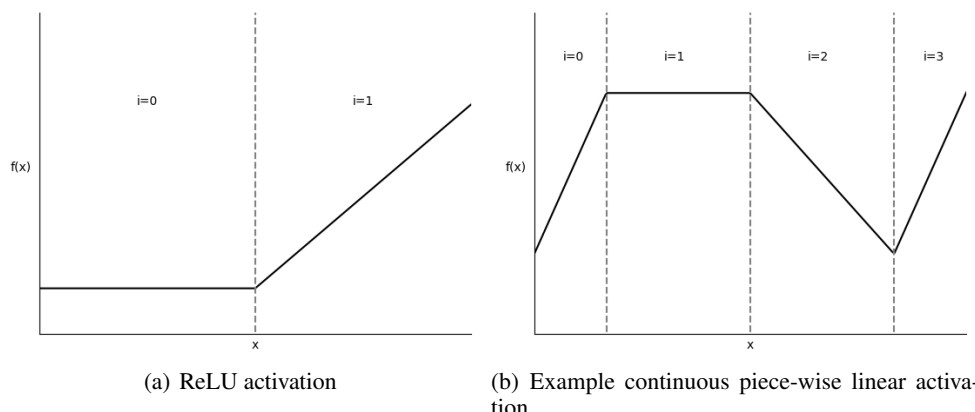

(a) ReLU activation

(b) Example continuous piece-wise linear activation

Figure 12: Visualizing the linear domains of the activation functions. Here we visualize $f(x)$ and $i$ such that we can use $a(i)$ in equation 25.

First, we assume the existence of a function $f(z)$ which computes both the activation function, as well as returning the linear domain label of the neuron activation. Additionally, we define a function $a(i)$ which returns the slope of the activation function given an activation label $i \in \mathbb{N}$. For a ReLU activation function, we define $f(z)$ and $a(i)$ in equation 25.

$$
\begin{aligned}
f(z) &= \begin{cases} (0,0) & \text{if } z < 0 \\ (z,1) & \text{if } z \geq 0 \end{cases} \\
a(i) &= \begin{cases} 0 & \text{if } i = 0 \\ 1 & \text{if } i = 1 \end{cases}
\end{aligned}
\tag{25}
$$

In order to derive the equivalent linear model given the activation configuration, we treat the entire networks as a series of linear transformations, which are then composed. First, we update the weights and biases to account for the slope of the activation function. Given weights $w$, biases $b$ and activation slopes $a$, we write the activation adjusted parameters for the i$^{\text{th}}$ neuron in the j$^{\text{th}}$ layer as:

$$
\begin{aligned}
w_{i,j}^* &= a_{i,j} w_{i,j} \\
b_{i,j}^* &= a_{i,j} b_{i,j}
\end{aligned}
\tag{26}
$$

From there, we can apply the linear transformations for each layer iteratively to obtain the equivalent linear transform. Given the weight $W$ and bias $B$ matrices, we iterate over the i$^{\text{th}}$ layers while accumulating the cumulative transformation:

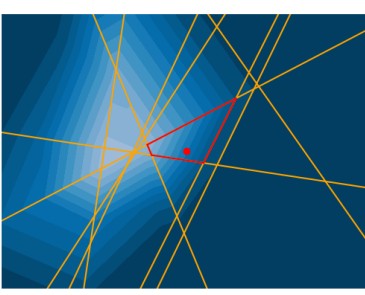 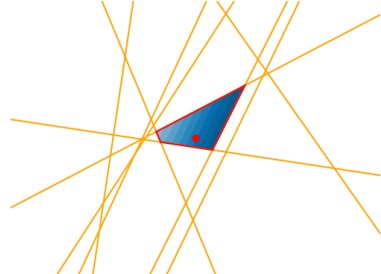

(a) Full ReLU Network output       (b) ReLU network equivalent linear model

Figure 13: For a given starting point (shown as a red dot), we compute the linear boundaries of the configuration using equation 28 (shown in orange), as well as the true linear domain (shown in red). Additionally, we show the equivalent linear model $\tilde{W}_i x + \tilde{B}$ within the linear domain.

$$\tilde{W}_i = \prod_{j<i} W_j^*$$

$$\tilde{B}_i = w_i^* \cdot B_{i-2}^* + B_{i-1}^* \tag{27}$$

When analyzing the boundaries of the linear domain, we note that an individual neuron is only non-linear when crossing the boundaries of the linear domains of the activation function. Consequently, for each individual neuron, given a lower bound $L_b$ and an upper bound $U_b$ bounding the current activation linear domain of said neuron, the following inequalities define the domain on which the neuron in linear:

$$L_b \leq w_{i,j} z_{j-1} + b_{i,j} \leq U_b \tag{28}$$

Furthermore, we note that the entire network remains linear as long as equation 28 holds for each neuron in the entire network, given the current configuration. Thus, by identifying the bounding inequalities of each neuron in the network and mapping it back to the input space, we identify the bounding inequalities that define the linear domain of the entire network with respect to a given activation configuration.

In practice, for each layer in the network, we identify the bounding inequalities for each neuron using equation 4.6. Next, we transform the linear inequalities back into the input space using the linearization of all the previous layers. Finally, we linearize the current layer using equation 4.4 and update the cumulative linearization mapping.

Subsequently, given the equivalent linear mapping $\tilde{W}$ and $\tilde{B}$, and the bounding inequalities of the domain expressed in the input space, we can define a quadratic programming sub-problem to compute the cSDF within the given linear domain. In the context of signed distance functions, we want to minimize the $L_2$ distance from a point $x_0 \in \mathbb{R}^N$ to the solution set of some given constraints. Since the $L_2$ norm is quadratic, we can rewrite the SDF of a linear model with linear constraints as a quadratic objective. Given a linear model $M(x) = Ax + b$, we impose the solid boundary $\partial S$ as an equality constraint of the form $Ax + b = k$, and derive the SDF quadratic objective in equation 32.

$$\begin{aligned}
||x - x_0||_2 &= (x - x_0)^T(x - x_0) \\
&= x^T x - 2x_0^T x + x_0^T x_0 \qquad \triangleright \text{ remove constant term} \\
&= x^T I x - 2x_0^T x \\
&= \frac{1}{2}(x^T 2Ix) + (-2x_0)^T x \\
&\therefore Q = 2I, c = -2x_0
\end{aligned} \qquad (29)$$

However, it is quite likely that a solution will not exist in the same linear domain as the starting point $x_0$. Consequently, we need to perform a local search of nearby cells to find solutions. In order to perform the local search, we need to efficiently enumerate neighbouring cells. From the previous discussion, we know that if a neuron switches activations, then it will lead to a different linear domain. As such, we can identify all neighbouring cells by exploring activation configurations that differ by exactly one neuron. In practice, not all configurations are guaranteed to exists, thus additional computation is needed to identify valid configurations. In particular, the neighbour enumeration step is accomplished by inverting the bounding inequality constraints one at a time, and subsequently checking if the configuration is feasible using linear programming. If the configuration is valid, only then is it explored.

Thus, we have now given mechanisms for finding solutions within a cell, as well as enumerating neighbouring cells given a starting cell. Consequently, we can combine them to form a local search algorithm for computing cSDFs of neural networks with continuous piece-wise linear activation functions. However, the choice of search algorithm has ramifications on the performance and theoretical correctness of the algorithm. For example, if we use a Breadth-First-Search (BFS) algorithm, then the first identified solution is guaranteed to be the closest. Unfortunately, BFS has exponential complexity in the average case, and thus is not suitable for large ReLU networks.

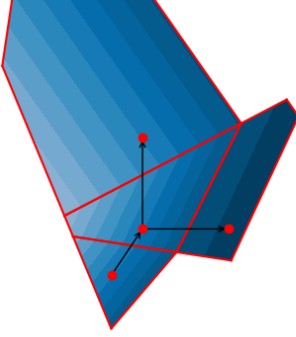

Figure 14: ReLU SDF bfs

Alternatively, search algorithms such as A* can perform much better in practice Zeng & Church (2009). In particular, A* performs a "best first" search, meaning that it explores cells based on a fitness function Hart et al. (1968). For the SDF algorithm the fitness function is a combination of distance and proximity to the constraint threshold:

$$\mathcal{L}(x_0, x, k, M) = \alpha||x - x_0||_2 + \beta|k - M(x)| \qquad (30)$$

By setting $\beta = 0$, the penalized heuristic will search the closest cells first, thus performing a breadth-first search. Alternatively, by setting $\alpha = 0$, A* will instead perform a greedy search. By choosing appropriate values of $\alpha$ and $\beta$, A* can balance between greedy and breadth-first search, thus finding the closest solution without the exponential scaling of breadth-first search. The overall algorithm for computing cSDFs of piece-wise linear neural networks is given in algorithm 2.

---

**Algorithm 2** Algorithm 2: A* SDF algorithm for piece-wise linear neural networks

---

**Require:** $W : (W_1, ..., W_n), W_i \in \mathbb{R}^{n_i \times m_i}, B : (B_1, ..., B_n), B_i \in \mathbb{R}^{m_i}, x_0 \in \mathbb{R}^n, k \in \mathbb{R}$
**Require:** $\alpha \in \mathbb{R}, \beta \in \mathbb{R}, f(z) : \mathbb{R} \to (\mathbb{R}, \mathbb{N}), a(i) : \mathbb{N} \to \mathbb{R}$
  $M = (W, B, f)$
  $\texttt{loss} \leftarrow \mathcal{L}(x_0, x_0, k, M)$
  $c \leftarrow \texttt{activation\_configuration(}x_0\texttt{,}M\texttt{)}$                                                      $\triangleright$ eq. 25-28, fig. 13, 14
  $\texttt{visited} \leftarrow \{c\}$
  $\texttt{queue} \leftarrow [(c, \texttt{loss})]$
  **while** $\texttt{length(stack) > 0}$ **do**
    $\texttt{loss}, c \leftarrow \texttt{queue.pop()}$
    **if** $c \notin \texttt{visited}$
      $\texttt{visited} \leftarrow \{\texttt{visited}, c\}$
      $\texttt{res} \leftarrow \texttt{linear\_program(}k, W, B, c\texttt{)}$
      **if** $\texttt{res} \neq \emptyset$
        **return** $\texttt{quadratic\_program(}k, W, B, c\texttt{)}$                         $\triangleright$ eq. 32
      **else**
        **for** $\nu \in \texttt{enumerate\_neighbours(}c\texttt{)}$
          **if** $\texttt{check\_valid(}\nu\texttt{)}$
            $x \leftarrow \texttt{get\_point\_in\_domain(}\nu\texttt{)}$
            $\texttt{loss} \leftarrow \mathcal{L}(x_0, x, k, M)$
            $\texttt{queue.push((loss, }\nu\texttt{))}$
        **end for**
  **end while**

---

## E  LOG-EXP-SUM CONTINUOUS BOOLEAN OPERATIONS

See figure 15 for hard max and Log-Exp-Sum iterations. In practice, the gradient interpolation leads to faster convergence in the multi-constraint setting. Additionally, we can characterize the error introduced by the approximation in equation 31. In fact, we can see that the smooth approximation has an error of at most $\frac{1}{\beta}\log(2)$. Since the $\frac{1}{\beta}\log 2$ is quite small, the approximation error is only meaningful when both values are close to zero. In addition, min and max operations can both be achieved by changing the sign of $\beta$, while changing the magnitude of $\beta$ affects the degree of smoothness when interpolating two SDFs.

$$\xi = e^{\beta \max(a,b)}$$

$$\nu = \frac{1}{\beta} \log \left(e^{\beta a} + e^{\beta b}\right)$$

$$= \frac{1}{\beta} \log \left(\xi + \eta\xi\right) \qquad \triangleright \quad 0 < \eta \leq 1$$

$$= \frac{1}{\beta} \log \left((1+\eta)\xi\right)$$

$$= \frac{1}{\beta} \left(\log(1+\eta) + \log(\xi)\right) \tag{31}$$

$$\nu = \frac{1}{\beta} \log(1+\eta) + \max(a,b)$$

$$|\nu - \max(a,b)| = |\frac{1}{\beta} \log(1+\eta)|$$

$$|\nu - \max(a,b)| \leq |\frac{1}{\beta} \log(2)|$$

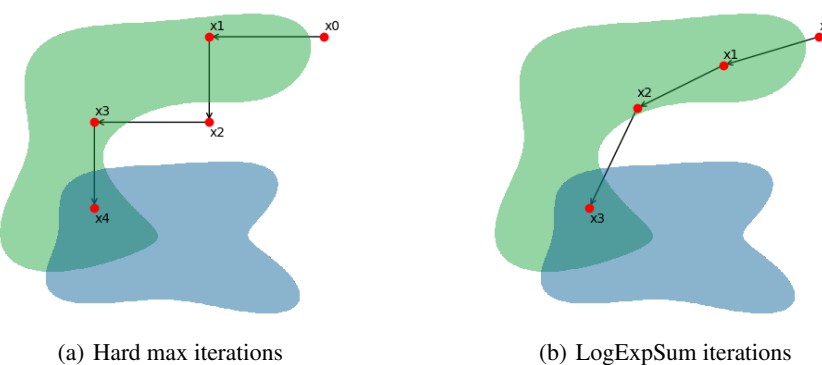

(a) Hard max iterations                    (b) LogExpSum iterations

Figure 15: Visualizing multi-constraint update steps using equation 11. When using the hard-max operation, we note that the iterations will take alternating steps optimizing one constraint at a time. Alternatively, using LogExpSum to approximate the max allows for smooth gradient interpolation, and a more direct optimization trajectory.

# F    MODEL ARCHITECTURES AND TRAINING DETAILS

## F.1    MNIST

**Auto-encoder model**  : The MNIST auto-encoder accepts flattened MNIST images as a 784-dimensional vector. The network has a latent size of 32. Sigmoid activation functions and a Xavier uniform weight initialization were used. The model was trained for 50 epochs using AdamW optimizer with a batch-size of 256 and a learning rate of $10^{-2}$. Mean Squared Error (MSE) was used as the loss function. See table 6 for network architecture.

Table 6: MNIST fully-connected auto-encoder architecture.

| Layer | Number of Neurons |
|-------|-------------------|
| 1 | 256 |
| 2 | 128 |
| 3 | 64 |
| 4 | 32 (latent dim) |
| 1 | 64 |
| 2 | 128 |
| 3 | 256 |
| 4 | 784 |

**Raw input SINN**  : The MNIST raw input SINN accepts flattened MNIST images as a 784-dimensional vector. The network has 100 nodes initialized from randomly selected training points. The model was trained for 25 epochs using AdamW optimizer with a batch-size of 256 and a learning rate of $10^{-4}$. MSE was used as the loss function.

**Auto-encoder SINN**  : The MNIST auto-encoder SINN accepts MNIST images encoded by the auto-encoder as a 32-dimensional vector. The network has 100 nodes initialized from randomly selected training points. The model was trained for 25 epochs using AdamW optimizer with a batch-size of 256 and a learning rate of $10^{-2}$. MSE was used as the loss function.

**Raw Input ReLU network**  : The Raw input ReLU network accepts flattened MNIST images as a 784-dimensional vector. The network architecture is given in table 7. The network uses ReLU activation functions and a Xavier uniform weight initialization. The model was trained for 25 epochs using AdamW optimizer with a batch-size of 256 and a learning rate of $10^{-4}$. Binary Cross Entropy with logit loss was used for the loss function.

Table 7: MNIST raw input ReLU architecture.

| Layer | Number of Neurons |
|-------|-------------------|
| 1 | 128 |
| 2 | 64 |
| 3 | 32 |
| 4 | 10 (output layer) |

**Auto-encoder ReLU network**  : The auto-encoder ReLU network accepts MNIST images encoded by the auto-encoder as a 32-dimensional vector. The network architecture is given in table 8. The network uses ReLU activation functions and a Xavier uniform weight initialization. The model was trained for 25 epochs using AdamW optimizer with a batch-size of 256 and a learning rate of $10^{-2}$. Binary Cross Entropy with logit loss was used for the loss function.

**ResNet oracle**  : The ResNet oracle was fine-tuned using a pretrained ResNet18 model with Imagenet 1k-V1 weightsPaszke et al. (2019). The fully connected portion of the network was replaced and retrained for 10 epochs, using AdamW optimizer with a learning rate of $10^{-2}$ and a batch

Table 8: MNIST auto-encoder ReLU architecture.

| Layer | Number of Neurons |
|-------|-------------------|
| 1 | 128 |
| 2 | 32 |
| 3 | 10 (output layer) |

size of 64. Cross-entropy loss was used for the loss function. The fine-tuned layer architecture is given in table 9. Notably, the fine-tuned layers include a 7x7 convolutional layer with stride=2 and padding=3.

Table 9: MNIST Oracle architecture.

| Layer | Layer Type | Number of Neurons |
|-------|-----------|-------------------|
| 1 | 7x7 conv | 64 |
| 2 | linear | 10 |

## F.2 CELEBA

**Auto-encoder model**   : The CelebA auto-encoder accepts flattened TinyVAE encodings as a 3136-dimensional vector. The network has a latent size of 256. ReLU activation functions and a Xavier uniform weight initialization were used. The model was trained for 100 epochs using Adam optimizer with a batch-size of 32 and a learning rate of $10^{-3}$. MSE was used as the loss function. The full network architecture is available in table 10.

Table 10: CelebA fully-connected auto-encoder architecture.

| Layer | Number of Neurons |
|-------|-------------------|
| 1 | 1500 |
| 2 | 1024 |
| 3 | 512 |
| 4 | 256 (latent dim) |
| 1 | 512 |
| 2 | 1024 |
| 3 | 1500 |
| 4 | 3136 |

**SINN architecture**   : The SINN accepts CelebA images encoded by TinyVAE and then the auto-encoder, as 256-dimensional vector. The network has 250 nodes initialized from randomly selected training points. The model was trained for 50 epochs using Adam optimizer with a batch-size of 32 and a learning rate of $10^{-2}$. MSE was used as the loss function.

**ReLU network**   : The ReLU network accepts CelebA images encoded by TinyVAE and then the auto-encoder, as 256-dimensional vector. The network architecture is given in table 11. The network uses ReLU activation functions and a Xavier uniform weight initialization. The model was trained for 50 epochs using Adam optimizer with a batch-size of 32 and a learning rate of $10^{-2}$. Binary Cross Entropy with logit loss was used for the loss function.

**ResNet oracle**   : The ResNet oracle was fine-tuned using a pretrained ResNet18 model with Imagenet 1k-V1 weightsPaszke et al. (2019). The fully connected portion of the network was replaced and retrained for 10 epochs, using Adam optimizer with a learning rate of $10^{-2}$ and a batch size of 32. Binary Cross-entropy with logit loss was used for the loss function. The a single linear layer was used as the fine-tuning layers for classification.

Table 11: CelebA raw input ReLU architecture.

| Layer | Number of Neurons |
|-------|-------------------|
| 1 | 128 |
| 2 | 64 |
| 3 | 40 (output layer) |

### F.3 ZINC

**SINN architecture** : The SINN accepts ZINC molecules encoded by ChemVAE as 196 dimensional vectors. The network has 250 nodes initialized from randomly selected training points. The model was trained for 25 epochs using Adam optimizer with a batch-size of 32 and a learning rate of $10^{-2}$. MSE was used as the loss function.

**ReLU network** : The ReLU network accepts ZINC molecules encoded by ChemVAE as 196-dimensional vectors. The network architecture is given in table 12. The network uses ReLU activation functions and a Xavier uniform weight initialization. The model was trained for 25 epochs using Adam optimizer with a batch-size of 32 and a learning rate of $10^{-2}$.MSE was used for the loss function.

Table 12: ZINC ReLU architecture.

| Layer | Number of Neurons |
|-------|-------------------|
| 1 | 128 |
| 2 | 64 |
| 3 | 3 (output layer) |

# G  CELEBA DETAILED RESULTS

Table 13: Test Accuracy and Input Size of models trained on CelebA.

| Model | Input Size | Test Accuracy |
|---|---|---|
| SINN | 256 | $86.00 \pm 0.07\%$ |
| ReLU | 256 | $85.90 \pm 0.07\%$ |
| Resnet | 224x224x3 | $85.85 \pm 0.07\%$ |

Table 14: Agreement Rates between the generated samples and the Resnet oracle for CelebA for single-constraint generation.

| Model | Agreement Rate |
|---|---|
| SINN | $52.00 \pm 2.67\%$ |
| ReLU | $54.48 \pm 2.11\%$ |

Table 15: Agreement rates between generated samples and the Resnet oracle for Celeba multi-constraint generation

| Model | Agreement Rate |
|---|---|
| SINN | $60.9 \pm 10.1\%$ |
| ReLU | $46.2 \pm 6.9\%$ |

Table 16: SDF Algorithm Runtime by model on CelebA dataset.

| Model | Runtime (in seconds) |
|---|---|
| SINN | $24.11 \pm 0.53$ |
| ReLU | $3.14 \pm 0.20$ |

Table 17: Individual class agreement rates for the CelebA dataset by model type

| Class | ReLU Agreement | SINN Agreement |
|---|---|---|
| Arched Eyebrows | $61.02 \pm 4.49\%$ | $64.00 \pm 6.79\%$ |
| Blond Hair | $71.88 \pm 5.62\%$ | $70.00 \pm 6.48\%$ |
| Brown Hair | $10.26 \pm 3.44\%$ | $24.00 \pm 6.04\%$ |
| Black Hair | $58.62 \pm 5.28\%$ | $44.00 \pm 7.02\%$ |
| Male | $73.48 \pm 3.84\%$ | $84.00 \pm 5.18\%$ |
| Pale Skin | $2.33 \pm 2.30\%$ | $16.00 \pm 5.18\%$ |
| 5 o'clock Shadow | $80.56 \pm 6.60\%$ | $62.00 \pm 6.86\%$ |

Table 18: Agreement Rates by model type between the guided gradient descent generated samples and the Resnet oracle for single-constraint CelebA.

| Model | Constraint Type | Agreement Rate |
|---|---|---|
| SINN | Single | $3.90 \pm 0.5\%$ |
| Relu | Single | $1.10 \pm 0.3\%$ |
| SINN | Multi | $0.00 \pm 0.0\%$ |
| Relu | Multi | $0.00 \pm 0.0\%$ |

# H MNIST DETAILED RESULTS

Table 19: Test Accuracy and Input Size of Models trained on MNIST.

| Model | Input Size | Test Accuracy |
|---|---|---|
| AE SINN | 32 | $91.72 \pm 0.28\%$ |
| AE ReLU | 32 | $92.87 \pm 0.26\%$ |
| Raw SINN | 784 | $83.27 \pm 0.37\%$ |
| Raw ReLU | 784 | $96.36 \pm 0.19\%$ |
| Resnet | 784 | $98.83 \pm 0.11\%$ |

Table 20: Agreement Rates by model type between the generated samples and the Resnet oracle for MNIST.

| Model | Agreement Rate |
|---|---|
| Raw SINN | $98.70 \pm 0.3\%$ |
| AE SINN | $100.00 \pm 0.0\%$ |
| Raw ReLU | $10.50 \pm 1.0\%$ |
| AE ReLU | $43.90 \pm 1.5\%$ |

Table 21: SDF Algorithm Runtime by model on MNIST dataset.

| Model | Input Size | SDF algorithm Runtime (in seconds) |
|---|---|---|
| AE SINN | 32 | $0.075 \pm 0.003$ |
| AE ReLU | 32 | $1.958 \pm 0.057$ |
| Raw SINN | 784 | $0.682 \pm 0.021$ |
| Raw ReLU | 784 | $21.616 \pm 0.352$ |

Table 22: Agreement Rates by model type between the guided gradient descent generated samples and the Resnet oracle for MNIST.

| Model | Agreement Rate |
|---|---|
| Raw SINN | $97.40 \pm 0.5\%$ |
| AE SINN | $97.50 \pm 0.5\%$ |
| Raw ReLU | $10.00 \pm 9.4\%$ |
| AE ReLU | $21.70 \pm 1.3\%$ |

# I ZINC DETAILED RESULTS

Table 23: Test Error and Input Size of models trained on Zinc.

| Model | Input Size | Test MSE |
|---|---|---|
| SINN | 196 | $37.2 \pm 0.33 \ (10^{-4})$ |
| ReLU | 196 | $44.3 \pm 0.37 \ (10^{-4})$ |

Table 24: Validity Rates of the generated samples for Zinc for single-constraint and multi-constraint generation when using GGD.

| Target | Model | Validity Rate |
|---|---|---|
| QED | SINN | $37.48 \pm 0.75$ % |
| SAS | SINN | $28.20 \pm 0.56$ % |
| QED + SAS | SINN | $31.56 \pm 0.63$ % |
| QED | ReLU | $36.96 \pm 0.74$ % |
| SAS | ReLU | $35.76 \pm 0.72$ % |
| QED + SAS | ReLU | $36.28 \pm 0.73$ % |

Table 25: ZINC single-constraint SDF algorithm run times.

| Target | Model | Runtime (in seconds) |
|---|---|---|
| QED | SINN | $13.85 \pm 1.30$ |
| SAS | SINN | $24.30 \pm 1.14$ |
| QED | ReLU | $11.04 \pm 0.49$ |
| SAS | ReLU | $2.74 \pm 0.50$ |

Table 26: Validity Rates of the generated samples for Zinc for single-constraint and multi-constraint generation.

| Target | Model | Validity Rate |
|---|---|---|
| QED | SINN | $44.72 \pm 1.26$% |
| SAS | SINN | $36.56 \pm 1.03$% |
| QED + SAS | SINN | $35.53 \pm 1.08$% |
| QED | ReLU | $35.60 \pm 1.01$% |
| SAS | ReLU | $38.00 \pm 1.07$% |
| QED + SAS | ReLU | $40.44 \pm 1.21$% |

Table 27: Zinc agreement rate for single and multi-constraint generation.

| Target | Model | Oracle | Agreement Rate |
|---|---|---|---|
| QED | SINN | Latent | $94.00 \pm 13.29$% |
| SAS | SINN | Latent | $86.00 \pm 12.16$% |
| QED + SAS | SINN | Latent | $93.02 \pm 14.19$% |
| QED | SINN | Analytical | $54.38 \pm 2.30$% |
| SAS | SINN | Analytical | $33.04 \pm 1.55$% |
| QED + SAS | SINN | Analytical | $15.18 \pm 0.78$% |
| QED | ReLU | Latent | $92.00 \pm 13.01$% |
| SAS | ReLU | Latent | $26.00 \pm 3.68$% |
| QED + SAS | ReLU | Latent | $20.00 \pm 2.98$% |
| QED | ReLU | Analytical | $46.74 \pm 2.22$% |
| SAS | ReLU | Analytical | $32.42 \pm 1.49$% |
| QED + SAS | ReLU | Analytical | $12.09 \pm 0.57$% |

Table 28: Zinc agreement rate for single and multi-constraint generation using GGD.

| Target | Model | Oracle | Agreement Rate |
|---|---|---|---|
| QED | SINN | Latent | $100.00 \pm 10.00$% |
| SAS | SINN | Latent | $71.00 \pm 7.10$ % |
| QED + SAS | SINN | Latent | $73.00 \pm 7.30$ % |
| QED | SINN | Analytical | $59.77 \pm 1.95$ % |
| SAS | SINN | Analytical | $29.79 \pm 1.12$ % |
| QED + SAS | SINN | Analytical | $15.46 \pm 0.55$ % |
| QED | ReLU | Latent | $97.00 \pm 9.70$ % |
| SAS | ReLU | Latent | $32.00 \pm 3.20$ % |
| QED + SAS | ReLU | Latent | $32.00 \pm 3.20$ % |
| QED | ReLU | Analytical | $58.98 \pm 1.94$ % |
| SAS | ReLU | Analytical | $27.74 \pm 0.93$ % |
| QED + SAS | ReLU | Analytical | $14.33 \pm 0.48$ % |

## J    MNIST GENERATED SAMPLES

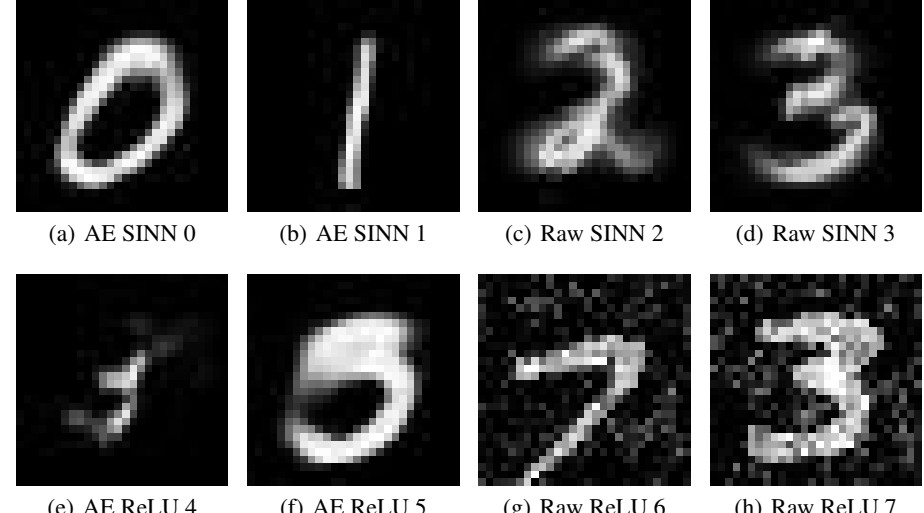

| (a) AE SINN 0 | (b) AE SINN 1 | (c) Raw SINN 2 | (d) Raw SINN 3 |

| (e) AE ReLU 4 | (f) AE ReLU 5 | (g) Raw ReLU 6 | (h) Raw ReLU 7 |

Figure 16: Sample generated MNIST samples using auto-encoder (AE) and input space (Raw) SINN and ReLU models.

## K CELEBA GENERATED SAMPLES

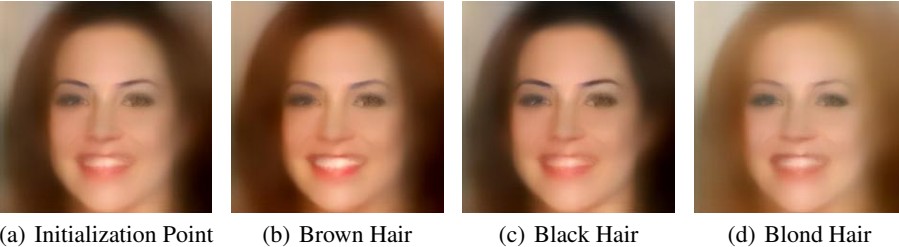

(a) Initialization Point     (b) Brown Hair     (c) Black Hair     (d) Blond Hair

Figure 17: Sample generated CelebA samples using SINN. The generated samples demonstrate the ability for composable constraints to edit attributes such as hair colour, while maintaining facial structure constant.

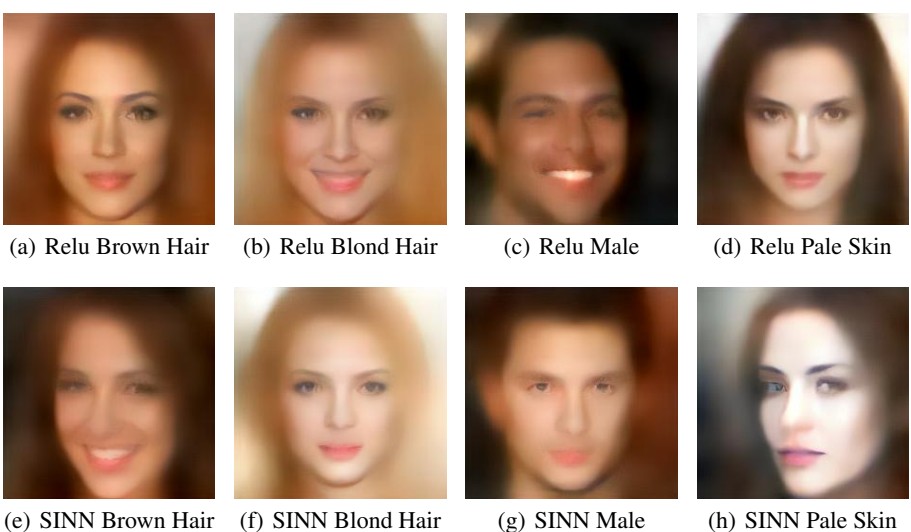

(a) Relu Brown Hair     (b) Relu Blond Hair     (c) Relu Male     (d) Relu Pale Skin

(e) SINN Brown Hair     (f) SINN Blond Hair     (g) SINN Male     (h) SINN Pale Skin

Figure 18: Sample generated CelebA samples using SINN and ReLU models for a variety of classes.

## L    ZINC GENERATED SAMPLES

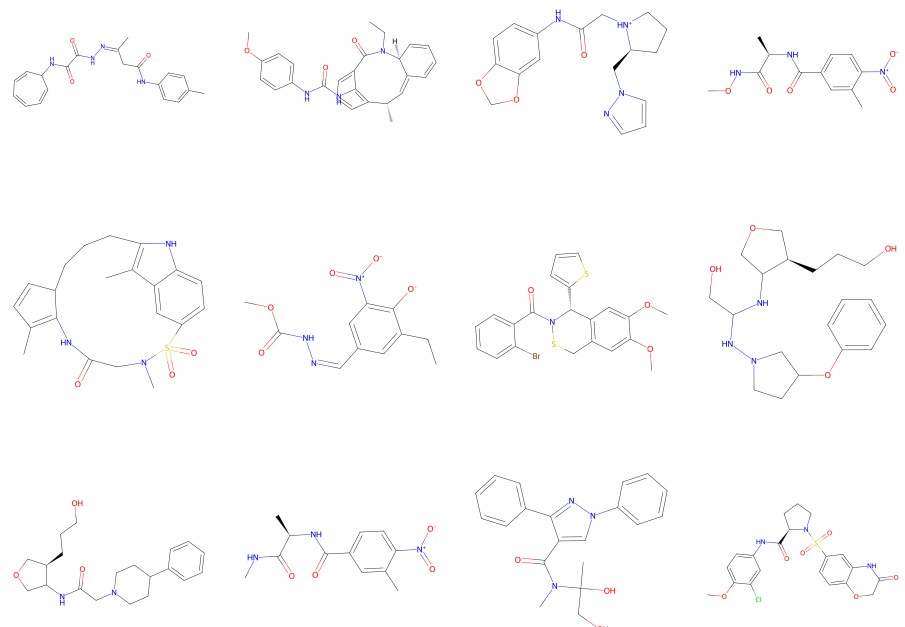

Figure 19: Sample generated chemical structures for the ZINC dataset.

## M    COMPUTATIONAL COMPLEXITY OF SDF ALGORITHMS

### M.1    SINN SDF COMPLEXITY

As can be seen from algorithm 1, computing the SDF requires enumerating the extrema of the function, as well as performing constrained optimization on each extrema. As shown in algorithm 1, augmented Lagrangians are used to compute the SDF. Augmented Lagrangians are a family of nonlinear constrained optimization techniques which only require computing the Jacobian, and thus scale linearly in the number of variables Nocedal & Wright (1999). Additionally, augmented Lagrangians scale linearly with the number of neurones in an SINN model. Thus, given $M$ neurons and $N$ dimensions, the computational complexity of Augmented Lagrangian is $O(MN)$. Additionally, we repeat the process for each of the $M$ neurons, for a computational complexity of $O(M^2N)$ for algorithm 1.

### M.2    ReLU SDF COMPLEXITY

As described in Algorithm 2, there are three major components of the computational complexity of the ReLU SDF: 1) the complexity of the local search algorithm (such as A*), 2) the complexity of solving linear programs for enumerating the regions and 3) the complexity of solving a quadratic program to obtain the actual SDF value.

In the context of signed distance functions, we want to minimize the $L_2$ distance from a point $x_0 \in \mathbb{R}^N$ to the solution set of some given constraints. Since the $L_2$ norm is quadratic, we can rewrite the SDF of a linear model with linear constraints as a quadratic objective. Given a linear model $M(x) = Ax + b$, we impose the solid boundary $\partial S$ as an equality constraint of the form $Ax + b = k$, and derive the SDF quadratic objective in equation 32.

$$
\begin{aligned}
||x - x_0||_2 &= (x - x_0)^T(x - x_0) \\
&= x^Tx - 2x_0^Tx + x_0^Tx_0 \qquad \triangleright \text{ remove constant term} \\
&= x^TIx - 2x_0^Tx \\
&= \frac{1}{2}(x^T2Ix) + (-2x_0)^Tx \\
\therefore\ Q &= 2I, c = -2x_0
\end{aligned}
\tag{32}
$$

In general, solving non-convex quadratic programs is NP-Complete Vavasis (1990). However, if the problem is convex, it can be solved in polynomial time. Given $N$ variables, $m$ constraints and $L$ bits, the computational complexity of a quadratic program is $O(N^4L^2)$ Ye & Tse (1989). Since the objective from equation 32 is convex, we can efficiently compute the signed distance function. Furthermore, several software packages such as `CVXOPT` and `quadprog` implement quadratic programming solvers, typically using either interior point, active set or augmented langrangian based algorithms Caron et al. (2024). Additionally, Vaidya provides worst-case bounds for various Linear Programming algorithms Vaidya (1989). Given $m$ constraints, $N$ variables and a factor $L = \log_2(1 + \det_m ax) + \log_2 p + \log_2(m + N)$, three algorithms are provided with computational complexities of $O(m^{1.5}NL)$, $O((mN^2 + m^{1.5}N)L)$ and $O(m^3L)$.

The computational complexity of the A* is $O(m^d)$, where $d$ is the depth, and $m$ is the number of neighbours of a given cell. For a given neurons, there are at most 2 action domains adjacent to the current activation domain. Thus, a given activation configuration and $n$ neurons, there are at most $2n$ neighbouring configurations. Thus, for the search algorithm we obtain a complexity of $O(n^d)$. Furthermore, for $N$ input variables, and $M$ constraints, the linear problem is at best solved in $O(M^{1.5}NL)$. The number of constraints is twice the number of neurons, such that $M = 2n$, thus the linear sub-problem is solvable in $O(n^{1.5}NL)$. Finally, the quadratic sub-problem must only be solved once, in $O(N^4L^2)$ time. Thus, the overall complexity of the ReLU SDF algorithm is $O(n^dn^{1.5}NL + N^4L^2)$. Notably, the complexity is dominated by the $n^d$ term. However, with proper hyper-parameters, the depth $d$ required to find a solution is quite small. This is the key advantage of using local search to compute SDFs in this context, as finding the solution minimizing the distance necessarily limits the depth to a small local neighbourhood.

