# OpenReview forum: "Solving Composable Constraints for Inverse Design Tasks"
_ICLR.cc/2025/Conference — Submitted to ICLR 2025_

### Official Review · Reviewer_Yf5i · 2024-10-27

**Soundness:** 1
**Presentation:** 2
**Contribution:** 1
**Rating:** 1
**Confidence:** 5

**Summary:**

This paper proposed a new optimization approach for inequality constraints modeled as signed distance functions (SDFs). The authors demonstrated the composability of SDFs and algorithms for approximating SDFs for a specific family of ML models of SINNs and piecewise linear nets. Experimental results on image and molecule datasets demonstrated better results.

**Strengths:**

- Algorithms for computing SDFs for SINNs and piecewise linear were proposed with solid mathematical proof.

**Weaknesses:**

1. **The motivation behind the paper on using SDFs for inequality constraints is unclear**. As a special case of the gradient-based method, it is unclear why we should use SDFs to model the constraints instead of directly optimizing along the gradient direction (or the reverse direction, depending on the task) of the objective $M(x)$. For all experiments in the paper, the thresholding operation seemed unnecessary, as a high metric always indicates better performance (e.g., more confidence in classification, or better synthesizability). The direct approach is also "composable" by summing over the gradient terms for different objectives.

2. **The generative models used in the paper were outdated**. VAE-based models are clearly inferior to diffusion or flow-based generative models, which have achieved remarkably better generation quality. Many existing pre-trained generative models exist with available checkpoints (e.g. Stable Diffusion series, [1], and [2] for molecules) and almost every up-to-date approach for solving inverse problems can be effectively applied to them (including, but not limited to [1], [3], [4], [5]). It is unclear why the authors did not carry out experiments on these models.

3. **The experiments were small-scale and limited**. In addition to the small and outdated generative model, the experiments were limited to classification tasks. Even though the authors proposed the approach to solving inverse problems, the standard inverse problems in CV domains like image inpainting, superresolution, and deblurring (all experimented in previous work) were never addressed.

4. **The baselines compared were extremely limited**. Essentially only the guided gradient approach was compared as the baseline, despite the fact that wide range of available methods for solving inverse problems. For example, [1]&[3] are gradient-free methods, [4]&[5] are gradient-based methods, and [6] is an RLHF approach applicable to non-differentiable objectives. [5]&[7] also extended constrained generation to molecule baselines. However, none of these approaches were compared.

5. **The proposed approach of calculating SDF may be time- and computation demanding**. The approximation for SDF relies on an iterative algorithm and differentiation through such an iterative procedure to obtain the gradient information. Therefore, it is expected to be extremely computation-intensive and memory-demanding, as all intermediate results must be stored for backpropagation. However, sampling time was never analyzed or provided as empirical results in the paper.

6. **Results in Figure 3 were extremely poor**. It looks as if the classifier has been adversarially attacked instead of generating the desired digit classes. The desirable digits were not even generated in (a)-(d) and were hardly distinguishable in (g). The captions are also confusing. If the authors started with some fixed image, which one was the starting image? Which one was the generation from the baseline?

7. Constraints in model machine learning practice often rely on a pre-trained model as the evaluator, e.g., the CLIP score in [4]. It is not practical to limit the classifier to piecewise linear nets or SINNs. The iterative algorithm also makes it prohibitively expensive to scale up to large pre-trained models like CLIP, especially when backpropagation is needed through the whole iterative algorithm.

[1] Mardani, Morteza, et al. "A variational perspective on solving inverse problems with diffusion models." arXiv preprint arXiv:2305.04391 (2023).

[2] Song, Yuxuan, et al. "Equivariant flow matching with hybrid probability transport." arXiv preprint arXiv:2312.07168 (2023).

[3] Kawar, Bahjat, et al. "Denoising diffusion restoration models." Advances in Neural Information Processing Systems 35 (2022): 23593-23606.

[4] Liu, Xingchao, et al. "Flowgrad: Controlling the output of generative odes with gradients." Proceedings of the IEEE/CVF Conference on Computer Vision and Pattern Recognition. 2023.

[5] Ben-Hamu, Heli, et al. "D-Flow: Differentiating through Flows for Controlled Generation." arXiv preprint arXiv:2402.14017 (2024).

[6] Wallace, Bram, et al. "Diffusion model alignment using direct preference optimization." Proceedings of the IEEE/CVF Conference on Computer Vision and Pattern Recognition. 2024.

[7] Hoogeboom, Emiel, et al. "Equivariant diffusion for molecule generation in 3d." International conference on machine learning. PMLR, 2022.

**Questions:**

Besides the issues mentioned in the Weakness section, I have the following additional questions:

8. I do not understand the claim made in the paper that SDFs are computationally intractable. For a given evaluation function or a pre-trained evaluation model $M(x)$, the SDF can be easily calculated as $-\mathrm{sign}(M(x)-k)\\|M(x)-k\\|_2$ for the constraint $M(x)\ge k$, which does not require any iterative algorithm to calculate. One can always directly calculate the gradient with respect to this objective (or $-\mathrm{sign}(M(x)-k)\\|M(x)-k\\|_2^2$, for stability).

9. For the same reason, the gradient guidance baseline used in the paper seemed to have adopted an erroneous setting according to the objective in line 119-120. Such an objective is designed for equality constraints but not for inequality constraints. The objective mentioned in my previous question should be the correct objective instead.

---

> ### Author Response · Authors · 2024-11-23
> **Rebuttal and answering questions**
>
> Thank you for your detailed review. We would like to address some of your comments, as well as clarify some mis-understandings.
>
> * Q1: First, modelling constraints as SDFs provides several useful properties for constructing complex systems of constraints. For example, we could construct an objective such as $(M_a(x) \geq k_a) \text{ XOR } (M_b(x) \geq k_b) \text{ XOR } (M_c(x) \geq k_c)$. This system of constraints is trivially implemented and solved using SDFs, but is not easily expressed in conventional gradient-based schemes. Additionally, a higher threshold is not universally useful. For example, when designing anti-microbial peptides (AMPs), we would want the peptide to have a helical fraction in a well defined range with a lower and upper bound. Consequently, assuming that we want to unconditionally maximize a model output greatly limits applicability outside a narrow range of classification tasks.
> * Q8/Q9: You make the claim that you do not understand why SDFs are intractible and then you provide an example for easily computing an SDF. This comes from a misunderstanding of the SDF formulation in equation 4. The example you provide computes the error in **Output Space**, while the SDF is the distance in **Input Space**. Thus, it is not as trivial as measuring the difference between the target and actual values, but rather the distance in feature space. This is a much harder problem as it requires an inversion of the predictive model, making the problem intractable in general.
> * Q2: While modern generative models (such as diffusion models) have achieved impressive success in a variety of tasks, VAEs provide several useful characteristics in this setting. In particular, we demonstrate that the introduction of a VAE improves computational performance, as well as generation quality. This is not the case for models such as Flow-matching, as they do not perform dimensionality reduction and do not induce useful latent spaces which can be used to train a classifier.
> * Q3: The core focus of this paper is to provide a mechanism for inverting predictive models on arbitrary tasks with arbitrary constraints. As such, the experiments provided are to demonstrate that the theoretical formulation of constraints as SDFs can be used to perform semantic operations. In fairness, it should be possible to apply composable constraints to non-classification tasks such as inpainting or super-resolution. However, this would require additional work for formulating a partially-observed variation of the SDF equations, and thus is out of scope for this work which introduces SDF constraints for the first time.
> * Q4: While other methods exist for solving inverse design tasks, we found it difficult to identify methods with similar enough properties to allow for a fair comparison. For example, [1] does not allow for zero-shot addition of arbitrary classifiers, while [4] and [5] does not easily permit complex systems of constraints which include boolean statements or conditional constraints. Consequently, we use GGD as a baseline, as it allows for the most fair comparison, since it is a method which explicitly inverts a classifier for the generation, and can be used in a post-hoc manner in a wide variety of settings.
> * Q5: The claim that we need to differentiate through the SDF algorithm is false. The intuition for obtaining the gradient is that algorithm 1 finds the nearest boundary point, thus producing the gradient is trivial since we return $x_0 - p_1$ in addition to $||x_0 - p_1||_2$. Please see algorithm 1 for more detail. Additionally, we have added a section in appendix M giving greater detail on the computational complexity of the SDF algorithms.
> * Q6: The claim is that "Results in Figure 3 were extremely poor, It looks as if the classifier has been adversarially attacked..." This is an astute observation, as we included the figure to demonstrate how SDFs constraints are susceptible to adversarial attacks in certain settings. In fact, we note on page 8 that "the ReLU model is liable to generate adversarial samples...". Indeed, we present the figure to point out that increased predictive accuracy does not guarantee improved sample quality, and that ReLU models are particularly susceptible to adversarial attacks.
> * Q7: The comment about expensive backpropagation through the SDF algorithm is incorrect (as previous stated). Additionally, the comment that piecewise linear networks are overly limited is odd given that this definition encompasses the majority of neural network architectures as ReLU is a widely used activation function, and we impose limitations on the architecture, only the activations.
>
> We understand that this paper proposes an unusual use of SDFs and that the intuition may be difficult to grasp for reviewers not familiar with domain. As such, we provided additional context in the rebuttal, and would like to point to Appendix B for a detailed 2D walk through of constructing and composing SDFs.

---

> > ### Comment · Reviewer_Yf5i · 2024-11-26
> > **Response to Authors' Rebuttal**
> >
> > While I thank the authors for clarifying some points regarding the problem setup in this paper, most of my concerns remain unaddressed, with no additional baselines compared and no theoretical or empirical analyses on the time/memory complexity. Below, I list a few critical points.
> >
> > 1. Regarding the poor experimental results in Figure 3&16. I noticed that the authors put a considerable amount of text analyzing the behaviors and algorithms for ReLU-based models in Section 4.2 and the whole of Appendix D. If the authors would claim the analysis for ReLU as part of their contributions, the poor generative results and its liability to adversarial attack significantly limit its application. On the other hand, if the results for ReLU are simply demo cases, it is unclear why the authors still provide a significant amount of content on their algorithms.
> > Indeed, the adversarial effects also lead to another serious issue that I missed before. As all the current experimental results in this work relied on a newly trained classifier, and since significant adversarial behaviors were observed, I believe the classifier scores do not accurately reflect the generative quality. Indeed, **not even one generative metric was evaluated in this work** though the authors claim it to be a constrained generative framework. As an easy example, the FID scores for the MNIST can be calculated (as we know the ground truth class-conditioned digits); the PSNR or PIPS scores (see [1, 3, 4, 5]) should be calculated for image generation quality whose ground truth is unknown; the atom/molecule stability (see [2, 5]) should be calculated for molecule generation (note the *pentavalent* nitrogen in Figure 19!). Otherwise, one cannot be sure the improved scores in the constraints are truly due to guidance or simply adversarial attacks.
> >
> > 2. Regarding additional baselines. Diffusion or flow-based generative model on latent space has been well-explored since the latent diffusion model (LDM) in 2021 and has inspired many well-known image generation models, including the Stable Diffusion series. Zero-shot guided generation for pre-trained LDMs is known in previous work as *latent traversal* [8] and relies on gradient information. Furthermore, I noted the factual error in the overall rebuttal when the authors claim that the existing approach requires "retraining your expensive generative model." This is not true, as [1, 3, 4, 5] are **all zero-shot training-free methods**. Therefore, I do not see a solid reason why these papers cannot be adopted for the setup in this work or, conversely, why the proposed method was not applied to diffusion models (which I believe can be partially attributed to the high running cost, as further elaborate below). Furthermore, it is also unclear why the authors trained a separate reward model when existing more robust and well-acknowledged CLIP scores are readily available (see, e.g. [4]).
> >
> > 3. Regarding the time/memory complexity. To clarify, I was concerned about the time and memory complexity because I noticed the **highly complicated, non-parallelizable, and CUDA-incompatible** iterative algorithm in Alg 2. To my understanding, such an algorithm is highly unscalability to larger generative models or larger reward models, as was also noted by other reviewers. Unfortunately, none of the theoretical asymptotic or empirical time and memory complexities was provided.
> >
> > 4. Regarding the baseline setup. The authors clarified their own setup but did not clarify their setup for their GGD baseline. According to the only two-line description of the implementation, I still believe the authors **erroneously implemented the equality constraints** instead. Gradient-based methods have been proven empirically very powerful in [4, 5]. It is highly unlikely such a baseline on CelebA can have such a significant performance gap (note [4] also carried out experiments on CelebA).
> >
> > Therefore, I believe the authors' rebuttal has provided little helpful information to justify its contribution to the generative domain, and I will hold my score.
> >
> >
> > [8] Song, Yue, et al. "Latent traversals in generative models as potential flows." arXiv preprint arXiv:2304.12944 (2023).

---

### Official Review · Reviewer_cHoY · 2024-11-01

**Soundness:** 3
**Presentation:** 3
**Contribution:** 3
**Rating:** 5
**Confidence:** 2

**Summary:**

This paper introduces an SDF-based approach for solving composable constraints in inverse design tasks. The core idea lies in representing the constraints using SDF and then using gradient-based method to optimize. In order to tackle the computation of SDE, the solutions using SINN and with ReLU have been proposed. The method has been investigated on image datasets and molecule datasets, showing that it can effectively optimize towards the target design objectives.

**Strengths:**

1. The approach is well motivated, the overall presentation is clear, and the method is easy to follow.

2. The proposed method of computing SDF using SINN and with ReLU activations is interesting and should be appreciated.

3. The method has been validated on image and molecule datasets.

**Weaknesses:**

1. While the approach is interesting, the scalability and practical applicability remain unclear. See Q1.

2. Lacking experiments on more complicated and practically meaningful inverse design tasks. See Q2.

3. Lacking important baselines like classifier (or classifier-free) guided diffusion models for inverse design. See Q3.

**Questions:**

Q1. What is the time/memory complexity of the algorithm using SINN/ReLU? Can the algorithm scale to larger datasets or bigger models?

Q2. Instead of the relatively preliminary inverse design tasks here presented in the paper, there are a lot more that are of practical interest, such as [1], [2]. Can the method be applied to these tasks?

Q3. The proposed approach can be viewed as a gradient-based approach. There are many other methods that directly build in inverse design targets in the generation process, such as using classifier-guidance or classifier-free guidance approaches with diffusion models. These methods should be considered as a valid baseline and would probably be more time-efficient than the proposed approach judging from the runtime provided in the appendix.

[1] Song et al. Solving Inverse Problems in Medical Imaging with Score-Based Generative Models. ICLR'22.

[2] Bao et al. Equivariant Energy-Guided SDE for Inverse Molecular Design. ICLR'23.

---

> ### Author Response · Authors · 2024-11-23
> **Rebuttal and answering questions**
>
> Thank you for your nice comments about the presentation and theory in the paper. Additionally, you raise several valid points in your questions and concerns:
>
> * We have some detailed analysis on the algorithmic complexity of both the SINN/Relu algorithms, and it seems like it was erroneously omitted from the appendices. In short, the SINN algorithm compute complexity is linear in the number of input dimensions and model parameters, and has a very small constant memory requirement. The ReLU SDF algorithm has exponential compute and memory requirements due to using a search heuristic (i.e. BFS/A*) and needing to maintain a stack. A more detailed explanation/proof is attached at the end of this comment. As such, the SINN algorithm is applicable to much larger datasets, while the ReLU algorithm would need some improvements to be practical for larger problems.
> * The restrictions for composable constraints are relatively mild. One only need a predictive model with an associated computable SDF. As such, given a suitable formulation of the problems described in [1], [2], composable constraints should be easily applicable. The main challenges we anticipate would be 1) collecting a dataset to suitable train a model and 2) formulating the labels in such a way that the consistency between multiple input sources are visible to the predictive model.
> * The core contribution of this paper is an interesting and novel mechanism for inverting a classifier under a system of constraints. While this methodlogy can be applied to other methods (such as diffusion models), this is somewhat outside of the scope of the claims being made.
> * Additionally, we find that direct comparisons are somewhat difficult to design. In settings with simple constraints, composable constraints are unfairly penalized as they perform extra compute in order to provide composability on objectives which are not composed. On the other hand, many existing methods cannot accommodate complex constraints such as $(M_a(x) \geq k_a) \text{ XOR } (M_b(x) \geq k_b) \text{ XOR } (M_c(x) \geq k_c)$, and so a comparison is not easy to construct. in fact, this is why we primarily chose to compare our method to guided gradient descent. GGD uses gradient information to invert a classifier given a system of constraints, and thus is a direct counterpart to composable constraints.
>
> Overall, our paper does not aim to outperform existing methods on existing baselines. Rather, we propose a novel mechanism for representing and solving systems of constraints, we then provide theoretical and empirical evidence that the formulation is valid.
>
> Additionally, models such as Diffusion Models, Flow-Matching, etc. are outside of the scope of this work. This work aims to provide a mechanism for efficiently inverting arbitrary classifiers with arbitrary constraints. Whether this method is applied directly on the data, on the latent space of a VAE or in a classifier-guided generation scheme is somewhat orthogonal to the computational and theoretical properties of inverting the classifier in the first place.

---

> > ### Comment · Reviewer_cHoY · 2024-11-28
> >
> > I thank the authors for the response.
> >
> > Though I acknowledge that the proposed method aims to invert arbitrary classifiers for arbitrary constraints, empirically I believe this is related to guidance methods widely used in generative models. Thus a comparison would be in need. I will keep my score.

---

### Official Review · Reviewer_hZoZ · 2024-11-03

**Soundness:** 2
**Presentation:** 3
**Contribution:** 2
**Rating:** 3
**Confidence:** 3

**Summary:**

In this paper, the authors propose representing inequality constraints in inverse design tasks via signed distance functions (SDF).  Furthermore, the authors provide two algorithms for computing SDF based on Shepard interpolation neural networks and piece-wise linear neural networks.

**Strengths:**

1. The paper is well-written and well-organized.


2. The research line for solving the constrained optimization through signed distance functions is interesting.

**Weaknesses:**

1.  The advantage of formulating the constrained optimization using the signed distance functions (SDF) compared with other constrained optimization solvers is not clear.

The authors employ the SDF to solve the constrained optimization. However, it is challenging to compute the boundary of the solution set in SDF (in Eq. (4)).  As a result, it can be very expensive to compute the SDF for general solution set S.

For the piece-wise linear neural networks,  the constrained optimization may be transformed as a linear constrained optimization problem.   What is the advantage of using SDF over linear programming solvers?  In addition,   even for piece-wise linear neural networks, the number of extreme boundary points grows exponentially as the depth grows.   The proposed method uses local search as an approximation. What is the advantage of the proposed method compared with other approximated solvers?   Because there is no comparison with other solvers and approximation techniques, it is unconvincing to justify the advantage and effectiveness of the proposed method.

Moreover, for the Shepard interpolation neural networks, the architecture may be too simple to handle the complex nature of the modern tasks in inverse design.

2.  The empirical evaluation is unconvincing without comparing with related constrained optimization techniques and baselines.

In the experiments,  it seems only one baseline is compared.  The comparison with related constrained optimization and other inverse design baselines (e.g., target generation techniques,  (off-line) black-box optimization techniques, etc. ).  Because of the missing comparison, it is unconvincing to justify the advantage and practical performance of the proposed method in inverse design.

Moreover, the experiments on MNIST and CelebA are two simple.  It is better to include more practical generation tasks, e.g., high-resolution (1024x1024) image tasks with more complex target properties for better evaluation.

**Questions:**

1.  What is the advantage of formulating the constrained optimization using the signed distance functions (SDF) compared with other constrained optimization solvers?  Please add a more detailed discussion and comparison with other constrained optimization solvers and approximated solvers.

2.  In the experiments, the empirical evaluation is unconvincing without comparing with related constrained optimization techniques and baselines.  Please add more comparisons with related constrained optimization methods and inverse design methods to justify the advantage of the proposed method.  In addition,  please include an evaluation of high-resolution (1024x1024) image tasks with more complex target properties to justify the practical performance of the proposed methods.

---

> ### Author Response · Authors · 2024-11-23
> **Rebuttal and answering questions**
>
> Thank you for your detailed review, you raise several interesting points. First, to address the mentioned weaknesses of the paper:
>
> * In the SINN case, the SDF algorithm is actually quite performant, and as described in the paper, can be computed in linear time. We added a revision which expands futher on the algorithmic complexity in Appendix M. However, you are correct that in general, the SDF algorithm may be expensive (such as the case for the ReLU SDF algorithm). Consequently, there is a fundamental tradeoff between the complexity of the model class, and the performance of SDF algorithm. Additionally, we note in the results section that the increased accuracy of a Resnet-based model doesn't necessarily translate to increased generation quality. Furthermore, while we demonstrate practical utility of SDFs for constraint satisfaction, we recognize the necessity to extend this approach to wider families of models to better understand the interactions between models families, network architectures, and generation quality. Such work would be highly technical and theoretical, and thus would be better places in subsequent work.
> * Regarding your question about the advantages of local-search versus other approximated solvers, I think that there may be a misunderstanding of Algorithm 2. In general, it is possible to extract the piecewise linear regions of a ReLU network and perform linear optimization. However, as described on page 6 "recent results from tropical geometry indicate that the number of linear domains grows combinatorially with the network depth, and as such it is intractable to enumerate all the domains". Thus we cannot use linear programming as-is, since the number of domains is much too large. In our paper, local search is not a replacement for linear programming, but an extension. We start with the piecewise linear region for the initial point, and then perform linear programming on each of the neighbouring domains. Thus we are not replacing or competing with linear programming solvers, but are using them as a component of a larger search strategy.
> * While composable constraints have not been shown to outperform existing methods on existing benchmarks, they provide interesting capabilities not possible with current methods. In particular, we can perform complex, post-hoc and zero-shot constrained generation, with the ability to incorporate multiple predictive models as needed. This is of particular interest in settings where the types of constraints may not be known ahead of time, such as in the case of chemical structure design for industrial chemicals. The rapidly shifting needs would not be well suited to the relatively static generative models which are popular today, and thus a post-method such as composable constraints would provide greater flexibility and tangible benefits in industrial settings.
>
> Overall, I think it would not be unreasonable to view the main concerns with this paper as being related to comparisons to SOTA methods. Your review asks for more detailed comparisons to existing methods, and larger datasets.
>
> While we believe that ZINC-250k and CelebFaceA (224x224) generation are sufficiently scaled-up, we cannot deny that larger datasets exist.
>
> However, we will push-back on the request for more detailed comparisons, as we believe it to be very difficult to construct a fair apples-to-apples comparison. In particular, composable constraints allow for complex boolean predicates to be built up. For example, it would not be unreasonable to have an objective such as $(M_a(x) \geq k_a) \text{ XOR } (M_b(x) \geq k_b) \text{ XOR } (M_c(x) \geq k_c)$. No current methods are able to solve constraints of this form. Thus, we either need compare composable constraints on simpler objectives, which it will be unfairly penalized due to the extra computation to provide the composable properties, or we could simply state that composable constraints work in settings where other models can't be used (and then be penalized for a lack of comparisons).
>
> We compare composable constraints to GGD, since it is the most similar method by virtue of accommodating flexible constraints and explicitely inverting a classifier.
>
> We would urge the reviewers to reconsider this paper in the context of a  paper that is proposing a novel methodology from first principles and providing evidence that it functions as intended, with the opportunity to extend the framework to incorporate cutting edge models once the theoretical underpinnings have been established.

---

> > ### Comment · Reviewer_hZoZ · 2024-11-29
> >
> > Thanks for the authors' response.  However, I am still concerned about the feasibility of the proposed method beyond the simple shallow network architecture.  Moreover,  many guided diffusion-model generation works have demonstrated strong performance for practical and challenging tasks, 1024x1024-sized images.  However, none of the comparisons with these works is unconvincing to justify the claim.

---

### Official Review · Reviewer_iyF8 · 2024-11-03

**Soundness:** 3
**Presentation:** 3
**Contribution:** 2
**Rating:** 5
**Confidence:** 4

**Summary:**

The paper introduces an approach based on sign distance functions (SDFs) to solve inverse design problems. The goal is primarily to find the closest feasible solution to any given starting point. The main selling point of SDFs in this context is (approximate) composability. A search algorithm is introduced to represent SDFs for composable constraints based on multi-valued predictors (SINNs or ReLU networks). SDF composition is softened using logsumexp for effective iterative corrections. The method is illustrated on simple problems involving MNIST, Celeba, and molecular design. The only empirical comparison is to a gradient guided design.

**Strengths:**

SDFs seem like reasonable ways of representing constraints, especially since, if correctly represented, a single adjusted gradient step would get you to a feasible point (when SDF is differentiable). SDFs can be composed across inequality constraints via max or min operations resulting in a bound which also seems reasonable. Search algorithms are introduced to approximately recover SDFs for predictors represented by SINNs or ReLU networks.

**Weaknesses:**

Many technical details are missing from the main text (relegated to the appendices). The paper should be rewritten by incorporating technical details back into the main text (the portion of the appendix that I read is clearer than the main text that leaves many questions unanswered).

The task that the paper addresses can be viewed as a general multi-criteria optimization problem to which there are numerous methods available. E.g., how would a simple linearization as a way to search for points on the pareto front work in comparison? This would be a variant of GDD but with different weights and would likely match better with the (greater?) computational requirements for SDFs. The proposed approach also relies on a low dimensional well-behaving latent space so that a SINN or ReLU network remains suitable for extrapolation and mapping to properties. It would be helpful to provide comparisons to methods that are not limited in this way. E.g., a conditional generative model could already be superior as it would not be forced to use a low dimensional latent space prior to generation and evaluation (by an oracle not operating on the latent space). At minimum, scaling to higher dimensional latent spaces seems essential to demonstrate. Also, if composability is the key argument, one should demonstrate this with more constraints (more than 2 or 3) and intersections. In the evaluation, one could control whether the constraints are largely aligned vs competing to see how the method (and baselines) succeed in different scenarios. Broadly speaking, the empirical results/comparisons should refer to the state of the art methods and performances in each chosen task. None of the example problems fit this description.

**Questions:**

Eq (2) does not seem to be used in the paper, only the feasibility problem from eq (1). Please clarify.

Theorem 1: please define "search algorithm" more precisely. Does it include augment gradient steps that start from extrema/critical points? What if no feasible solution exists?

Algorithm 1: x0 is presumably just x as in the rest of the algorithm. Algorithm here is defined for one dimensional x. Please rewrite. The algorithm also only returns a distance as a value while the design problem requires a gradient of the SDF. How is the gradient calculated on the basis of Algorithm 1?

How is the closeness to the initial point controlled or assessed when searching for a feasible point? Or is the assessment global?

---

> ### Author Response · Authors · 2024-11-23
> **Rebuttal and answering questions**
>
> Thank you for the detailed review, we are glad to see that the idea of representing constraints as SDFs is received positively.
>
> Now to address the two main concerns with the paper:
>
> * We agree that much of the technical content is relegated to the appendix. However, we explicitly point the reader to the relevant sections in the appendix when applicable (for example on page 5 "The full details are given in algorithm 1 and appendix C."). Unfortunately, the ICLR format only permits 10 pages in the main text and we believe that all of the content in the main body is essential, however we would be grateful for any specific recommendations of section(s) that we could remove or trim.
> * We would like to point at the composable constraints do not depend on semantic latent-spaces as such. While relatively small scale, we demonstrated composable constraints in raw-input spcace in the MNIST experiments. The addition of VAEs for the CelebFace A dataset is mostly due to numerical stability and lack of available solvers for augmented lagrangians. However, we do note that the introduction of dimensionality reduction techniques is advantageous in terms of solution quality and runtime. Finally, simple linearization may work, but suffers from a lack of principled methods for balancing quality of solution versus proximity to the initial point. One could construct objectives such as $\mathcal{L}(x) = \alpha\cdot(M_i(x) - y^*)^2 + \beta\cdot||x_0 - x||_2$, but this would open a separate slew of challenges of hyper-parameter tuning and soft versus hard constraint satisfaction. Thus, we believe that this comparison is not trivial.
> * Thank you for your comments on the clarity of algorithm 1. We added an updated version to address the typographical error, as well as to show how the gradient is computed. The intuition for obtaining the gradient is that algorithm 1 finds the nearest boundary point, thus producing the gradient is trivial since we can return $x_0 - p_1$ alongside $||x_0 - p_1||_2$
> * The closeness of the solution to the initial point is fundamentally tied to the SDF formulation and eqs. 10 and 11. Since the SDF function (and gradient) always point to the nearest solution, then any obtained solution will necessarily be the nearest solution. Thus, the solution produced is a point which is as close as possible to the initial point, while still satisfying the system of constraints.
>
>
> For your questions:
>
> * Indeed Eq. 2 is not used in the paper, but rather to make a clear distinction and statement of constrained optimization and inverse design.
> * In this context, we use the term "Search Algorithm" to denote an algorithm which will somehow search the input space for solutions to the system of constraints. Given the structure of an SINN, we know that the nearest solution will exist on the boundary between the starting point and the SINN nodes. This allows us to provably obtain the minima in linear time (as opposed to a more naive algorithm such as grid search). The augmented gradient step is used as a particular implementation of this theorem. We have added an additional section in the appendix describing the computational complexity of the SDF algorithms (See appendix M).
> * The question about feasible solutions is quite astute and we want to address this in more detail. In the single constraint case, it is possible to analytically detect whether a feasible solution exists. As such, if a single constraint is not satisfiable, we can detect this in O(1) time. In the multi-constraint case, this is not possible and we would instead need to either limit the number of iterations of eq. 11, or implement an early stopping mechanism that would detect whether the solution is improving over time.

---

> > ### Comment · Reviewer_iyF8 · 2024-11-24
> >
> > Thank you for the clarifications. I would like to point out a few additional things:
> > - the latent space dimension seems relevant since the authors need the predictors to be simple (SINN or small ReLU network). No such requirement applies to GDD as it could accept any predictor. It would be nice to see how a GDD with a more accurate predictor compares with the authors approach.
> > - the authors' method does involve hyperparameters. E.g., beta used to soften composition of SDFs; learning rate since a single step (esp when composing) will not get you to a satisfactory solution. What is the single constraint single step success rate in your case? A stopping criterion for the gradient steps is also needed, similar to GDD.
> > - GDD costs should not be two-sided as stated. There's no problem finding a better solution wrt one constraint in an attempt to also satisfy another. A proximal point optimizer with respect to a couple of constraints, for example would require only two method specific hyper-parameters (relative weight of the constraints together with the weight of the proximal penalty)
> > - your tables should also report min/max/ave distance to the initial point for each method since this is explicitly used as a motivation
> > - it would be helpful to illustrate the method itself further in many ways since the example problems are toy problems. E.g., quality of gradients in the multi-criteria setting across different constraints, how many iterations are needed to find a satisfying solution, scalability in terms of latent dimension, SINN vs more complex predictor quality, etc.
> > - feasibility is a real issue in applications (e.g., in drug design you would not know whether a target solution exists). A discussion of how this is handled would be important to include in the paper.

---

> > > ### Author Response · Authors · 2024-11-24
> > > **Additional Context/Information**
> > >
> > > Thank you for your comment:
> > >
> > > * I would like to point out the difference between the requirements of the composable constraint framework, and the practical methods presented in the paper. In general, composable constraints only require a computable SDF function, which is quite permissive. However, in addition to the composable constraint framework, we present two practical algorithms for computing SDFs for concrete families of models. The main limitation here is additional work to derive efficient SDF algorithms for broader families of models, which is somewhat out of scope and further work would essentially be able to build up a model/algorithm zoo for diverse collections of model families.
> > > * There seems to be a slight misunderstanding of the method in the single-constraint case. Due to the formulation of the SDF, the success rate for the single-constraint case (as measured by the classifier itself) is 100%. Put otherwise, if a solution exists, then the single-constraint case will *always* produce a valid inversion of the predictive model. Additionally, since the SDF represents membership to the solution region, it is trivial to detect when a solution has been obtained. Furthermore, there is not an issue of vanishing gradient, as the objective always has a gradient of magnitude 1, thus we do not need to take the same amount of care to define things like step sizes and early stopping criterion.
> > > * To reiterate on the differences between GGD and composable constraints, there are a few points I'd like to emphasize. 1) GGD does not provide a principled way of handling compositional or conditional constraints (i.e. the XOR example previous described). Yes, one can construct objectives in an ad-hoc way, subject to additional hyper-parameter tuning, but this need be done on a case-by-case basis. 2) in the single-constraint case, GGD typically requires many steps and tuning of the step size and early stopping criterion, while composable constraints can produce a solution in a single step. Overall, given a specific setting, it is likely possible to construct a GGD scheme for that setting (given various additions as mentioned), but this would be both an ad-hoc approach, as well as significant extensions to GGD. In which case, another advantage of composable constraints is that you don't need to produce an ad-hoc scheme or extensions for each use-case, but that you have a single unified approach which can always be applied.
> > > * Thank you for the point about adding the min/max/mean distance of the solutions to the starting point, this would definitely help strengthen the case for composable constraints and SDFs.
> > > * As mentioned, in the case of the SINN, it is possible to detect feasibility in the single constraint case efficiently. In the multi-constraint case (or when using more complex models), feasibility is much harder to detect pre-runtime. However, this is a weakness shared by many other generative and inverse design methodologies. In the future, it may be possible to either construct accurate models which allow for analysis of feasibility, or make headway in analyzing existing models (such as ReLU networks). However, this is likely closely related to model explainibility, which is known to be a hard problem, and thus would be out of scope for the initial work introducing the methodology.

---

> > > > ### Comment · Reviewer_iyF8 · 2024-11-24
> > > >
> > > > To clarify: the authors' approach requires a simple predictor (e.g., SINN). A single step success with respect to this is indeed 100% but not in terms of the oracle. Calculating the single step success of a single constraint as in the tables (wrt oracle) is another way of measuring how much one loses by having to adopt a simple predictor so that SDFs can be calculated. GDD should not be required to use limited predictors.

---

### Author Response · Authors · 2024-11-23
**Overall Rebuttal**

In this work, we focus on introducing a novel methodology. The experiments are designed to show that the method does in fact work, even in higher dimensions, and as well or better than the most commonly-used alternative. As such, we are not trying to demonstrate better performance on a specific task. Instead, we propose a novel theoretical framework and provide theoretical analysis and empirical validation that the framework is correct and that the methodology performs useful operations when applied to real settings. None of this is supposed to outperform any given model on any specific baseline, but rather all in service of providing evidence that the theoretical analysis is sound.

Several reviewers mention the use of more sophisticated generative models (such as Flow-Matching or Diffusion models). However, when using such models, changing the constraints (or types of constraints) to solve would require retraining your expensive generative model. We want to stress that composable constraints are a flexible post-hoc method that allows for interactive modification of the types of constraints in a zero-shot manner.

Finally, we are explicitly trying to invert a predictive model, and so the comparison to generative models is not quite apt. Generative models are trained specifically to generate samples, while composable constraints are intended to take an existing classifier and invert it. In fact, we use GGD as a baseline specifically since it is a method typically used to invert classifiers. Consequently, pure generative models that do not induce a semantic latent space (such as Diffusion models) are not directly applicable and would require further work to integrate with composable constraints.

---

### Meta-Review · Area_Chair_LXX1 · 2024-12-25

**Metareview:**

The paper proposes Signed Distance functions (SDFs) as  a way to represent constraints. SDFs allow to get a feasible point with a single adjusted step, and can be composed  across multiple inequality constraints using min or max operations. Authors provide  algorithms for computing SDF based on Shepard interpolation neural networks and piece-wise linear neural networks.

The paper is interesting and has good potential nevertheless reviewers mentioned multiple issues with the current drafts:
* it lacks on clarity as pointed out by the reviewers, many of the parts in the appendices should be incorporated in the main text to improve the clarity and the readability of the work.
* Going beyond toy examples in the experimentation
* Going beyond simple predictors to compute SDFs
* Explaining the choices of the hyperparameters and its impact on the success of finding a feasible point
* Adding the min/max/mean distance of the solutions to the starting point
* Guided gradient Descent is limited by the authors to simple predictors to compare to their method , which is a limiting setup

We encourage the authors to incorporate reviewers feedback and submit to the next venue.

**Additional Comments On Reviewer Discussion:**

The paper was discussed at length between authors and reviewers.

---

### Decision · Program_Chairs · 2025-01-22

Reject